# AUTOGEN: ENABLING NEXT-GEN LLM APPLICATIONS VIA MULTI-AGENT CONVERSATION

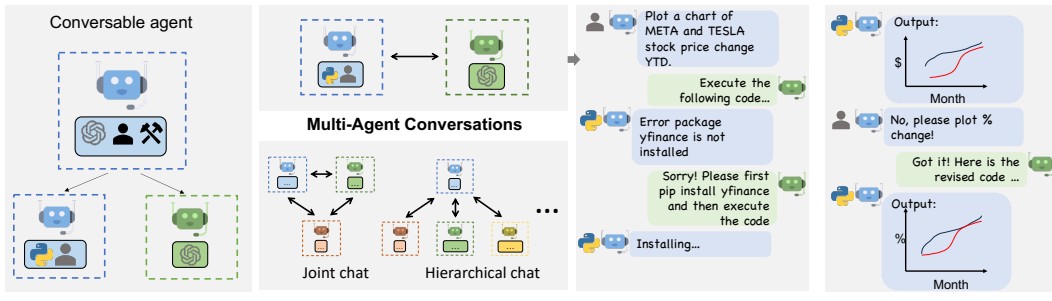

Figure 1: `AutoGen` enables diverse LLM-based applications using multi-agent conversations. (Left) `AutoGen` agents are conversable, customizable, and can be based on LLMs, tools, humans, or even a combination of them. (Top-middle) Agents can converse to solve tasks. (Right) They can form a chat, potentially with humans in the loop. (Bottom-middle) The framework supports flexible conversation patterns.

## ABSTRACT

`AutoGen`[1] is an open-source framework that allows developers to build LLM applications via multiple *agents* that can converse with each other to accomplish tasks. `AutoGen` agents are customizable, *conversable*, and can operate in various modes that employ combinations of LLMs, human inputs, and tools. Using `AutoGen`, developers can also flexibly define agent interaction behaviors. Both natural language and computer code can be used to program flexible conversation patterns for different applications. `AutoGen` serves as a generic infrastructure to build diverse applications of various complexities and LLM capacities. Empirical studies demonstrate the effectiveness of the framework in many example applications, with domains ranging from mathematics, coding, question answering, operations research, online decision-making, entertainment, etc.

## 1 INTRODUCTION

Large language models (LLMs) are becoming a crucial building block in developing powerful *agents* that utilize LLMs for reasoning, tool usage, and adapting to new observations (Yao et al., 2022; Xi et al., 2023; Wang et al., 2023b) in many real-world tasks. Given the expanding tasks that could benefit from LLMs and the growing task complexity, an intuitive approach to scale up the power of agents is to use multiple agents that cooperate. Prior work suggests that multiple agents can help encourage divergent thinking (Liang et al., 2023), improve factuality and reasoning (Du et al., 2023), and provide guardrails (Wu et al., 2023). In light of the intuition and early evidence of promise, it is intriguing to ask the following question: *how* can we facilitate the development of LLM applications that could span a broad spectrum of domains and complexities based on the multi-agent approach?

Our insight is to use *multi-agent conversations* to achieve it. There are at least three reasons confirming its general feasibility and utility thanks to recent advances in LLMs: First, because chat-optimized LLMs (e.g., GPT-4) show the ability to incorporate feedback, LLM-agents can cooperate

---

[1]`https://anonymous.4open.science/r/autogen-iclr2024`

through *conversations* with each other or human(s), e.g., a dialog where agents provide and seek reasoning, observations, critiques, and validation. Second, because a single LLM can exhibit a broad range of capabilities (especially when configured with the correct prompt and inference settings), conversations between differently-configured agents can help combine these broad LLM capabilities in a modular and complementary manner. Third, LLMs have demonstrated ability to solve complex tasks when broken into simpler subtasks. Multi-agent conversations can enable this partitioning and integration in an intuitive manner.

How can we leverage the above insights and support different applications with the common requirement of coordinating multiple agents, potentially backed by LLMs, humans, or tools exhibiting different capacities? We desire a multi-agent conversation framework with generic abstraction and effective implementation that has the flexibility to satisfy different application needs. Achieving this requires addressing two critical questions: (1) How can we design individual agents that are capable, reusable, customizable, and effective in multi-agent collaboration? (2) How can we develop a straightforward, unified interface that can accommodate a wide range of agent conversation patterns? In practice, applications of varying complexities may need distinct sets of agents with specific capabilities, and may require different conversation patterns, such as single- or multi-turn dialogs, different human involvement modes, and static vs. dynamic conversation. Moreover, developers may prefer the flexibility to program agent interactions in natural language or code. Failing to adequately address these two questions would limit the framework's scope of applicability and generality.

While there is contemporaneous exploration of multi-agent approaches,[2] we develop `AutoGen`, a generalized multi-agent conversation framework (Figure 1), based on the following new concepts.

1  **Customizable and conversable agents.** `AutoGen` uses a generic design of agents that can leverage LLMs, human inputs, tools, or a combination of them. The result is that developers can easily and quickly create agents with different roles (e.g., agents to write code, execute code, wire in human feedback, validate outputs, etc.) by selecting and configuring a subset of built-in capabilities. The agent's backend can also be readily extended to allow more custom behaviors. To make these agents suitable for multi-agent conversation, every agent is made *conversable* – they can receive, react, and respond to messages. When configured properly, an agent can hold multiple turns of conversations with other agents autonomously or solicit human inputs at certain rounds, enabling human agency and automation. The conversable agent design leverages the strong capability of the most advanced LLMs in taking feedback and making progress via chat and also allows combining capabilities of LLMs in a modular fashion. (Section 2.1)

2  **Conversation programming.** A fundamental insight of `AutoGen` is to simplify and unify complex LLM application workflows as multi-agent conversations. So `AutoGen` adopts a programming paradigm centered around these inter-agent conversations. We refer to this paradigm as *conversation programming*, which streamlines the development of intricate applications via two primary steps: (1) defining a set of conversable agents with specific capabilities and roles (as described above); (2) programming the interaction behavior between agents via conversation-centric *computation* and *control*. Both steps can be achieved via a fusion of natural and programming languages to build applications with a wide range of conversation patterns and agent behaviors. `AutoGen` provides ready-to-use implementations and also allows easy extension and experimentation for both steps. (Section 2.2)

`AutoGen` also provides a collection of multi-agent applications created using conversable agents and conversation programming. These applications demonstrate how `AutoGen` can easily support applications of various complexities and LLMs of various capabilities. Moreover, we perform both evaluation on benchmarks and a pilot study of new applications. The results show that `AutoGen` can help achieve outstanding performance on many tasks, and enable innovative ways of using LLMs, while reducing the development effort. (Section 3 and Appendix D)

---

[2]Note for reviewers: As of September 23rd, 2023, most prior multi-agent LLM approaches have not been published at peer-reviewed venues and only appeared online after May 2023. Hence, per the ICLR reviewer guide they should be considered as contemporaneous work. We refer to Appendix A for a detailed discussion.

## 2 THE AUTOGEN FRAMEWORK

To reduce the effort required for developers to create complex LLM applications across various domains, a core design principle of `AutoGen` is to streamline and consolidate multi-agent workflows using multi-agent conversations. This approach also aims to maximize the reusability of implemented agents. This section introduces the two key concepts of `AutoGen`: conversable agents and conversation programming.

### 2.1 CONVERSABLE AGENTS

In `AutoGen`, a *conversable agent* is an entity with a specific role that can pass messages to send and receive information to and from other conversable agents, e.g., to start or continue a conversation. It maintains its internal context based on sent and received messages and can be configured to possess a set of capabilities, e.g., enabled by LLMs, tools, or human input, etc. The agents can act according to programmed behavior patterns described next.

**Agent capabilities powered by LLMs, humans, and tools.** Since an agent's capabilities directly influence how it processes and responds to messages, `AutoGen` allows flexibility to back its agents with various capabilities. `AutoGen` supports many common composable capabilities for agents, including **1) LLMs.** LLM-backed agents exploit many capabilities of advanced LLMs such as role playing, implicit state inference and progress making from conversation history, providing feedback, adapting from feedback, and coding. These capabilities can be combined in different ways via novel prompting techniques[3] to increase skill and autonomy of an agent. `AutoGen` also offers enhanced LLM inference features such as result caching, error handling, message templating, etc., via an enhanced LLM inference layer. **2) Humans.** Human involvement is desired or even essential in many LLM applications. `AutoGen` lets a human participate in agent conversation via human-backed agents, which could solicit human inputs at certain rounds of a conversation depending on the agent configuration. The default *user proxy* agent allows *configurable* human involvement levels and patterns, e.g., frequency and conditions for requesting human input including the option for humans to skip providing input. **3) Tools.** Tool-backed agents have the capability to execute tools via code execution or function execution. For example, the default user proxy agent in `AutoGen` is able to execute code suggested by LLMs, or make LLM-suggested function calls.

**Agent customization and cooperation.** Based on application-specific needs, each agent can be configured to have a mix of basic back-end types to display complex behavior in multi-agent conversations. `AutoGen` allows easy creation of agents with specialized capabilities and roles by reusing or extending the built-in agents. The yellow-shaded area of Figure 2 provides a sketch of the built-in agents in `AutoGen`. The `ConversableAgent` class is the highest-level agent abstraction and, by default, can use LLMs, humans, and tools. The `AssistantAgent` and `UserProxyAgent` are two pre-configured `ConversableAgent` subclasses, each representing a common usage mode, i.e., acting as an AI assistant (backed by LLMs) and acting as a human proxy to solicit human input or execute code/function calls (backed by humans and/or tools). In the example on the right-hand side of Figure 1, an LLM-backed assistant agent and a tool- and human-backed user proxy agent are deployed together to tackle a task. Here, the assistant agent generates a solution with the help of LLMs and passes the solution to the user proxy agent. Then, the user proxy agent solicits human inputs or executes the assistant's code and passes the results as feedback back to the assistant.

By allowing custom agents that can converse with each other, conversable agents in `AutoGen` serve as a useful building block. However, to develop applications where agents make meaningful progress on tasks, developers also need to be able to specify and mold these multi-agent conversations.

### 2.2 CONVERSATION PROGRAMMING

As a solution to the above problem, `AutoGen` utilizes *conversation programming*, a paradigm that considers two concepts: the first is *computation* – the actions agents take to compute their response in a multi-agent conversation. And the second is *control flow* – the sequence (or conditions) under which these computations happen. As we will show in the applications section, the ability to pro-

---

[3]Appendix C presents an example of such novel prompting techniques which empowers the default LLM-backed assistant agent in `AutoGen` to converse with other agents in multi-step problem-solving.

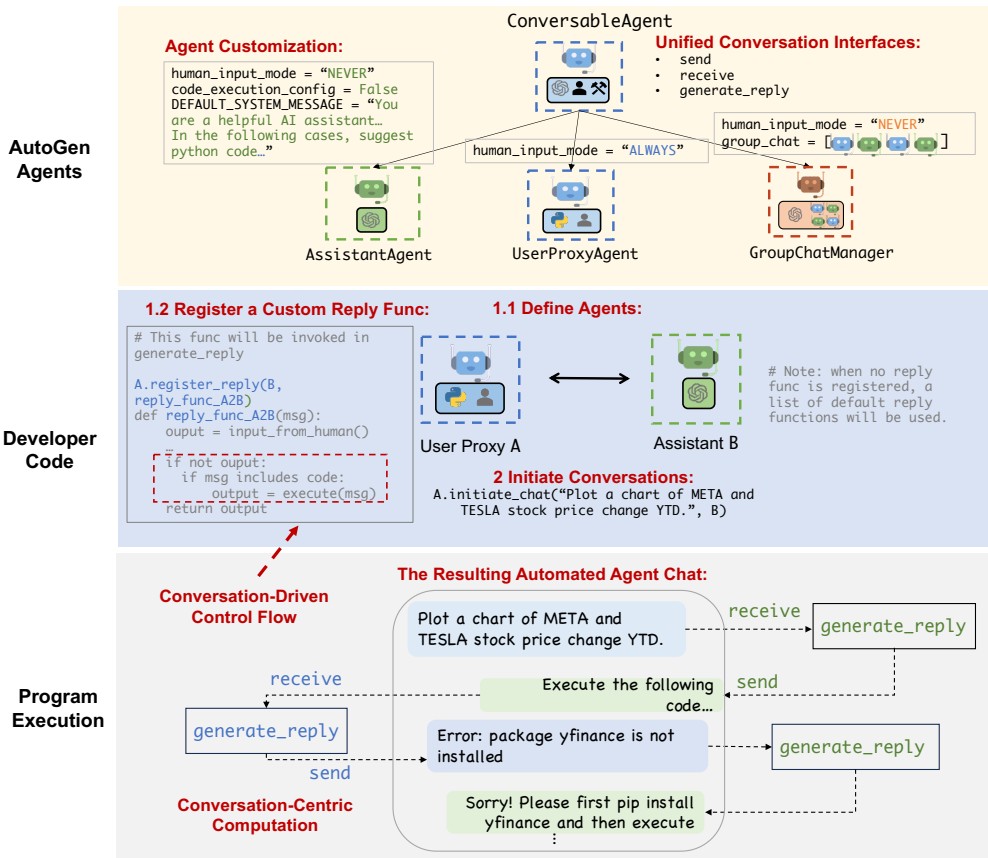

Figure 2: Illustration of how to use `AutoGen` to program a multi-agent conversation. The top sub-figure illustrates the built-in agents provided by `AutoGen`, which have unified conversation interfaces and can be customized. The middle sub-figure shows an example of using `AutoGen` to develop a two-agent system with a custom reply function. The bottom sub-figure illustrates the resulting automated agent chat from the two-agent system during program execution.

gram these helps implement many flexible multi-agent conversation patterns. In `AutoGen`, these computations are conversation-centric. An agent takes actions relevant to the conversations it is involved in and its actions result in message passing for consequent conversations (unless a termination condition is satisfied). Similarly, control flow is conversation-driven – the participating agents' decisions on which agents to send messages to and the procedure of computation is a function of the inter-agent conversation. This paradigm helps one to reason about a complex workflow as agent action taking and conversation message passing between agents intuitively.

Figure 2 provides a simple illustration. The bottom sub-figure shows how individual agents perform their role-specific, conversation-centric computations to generate responses (e.g., via LLM inference calls and code execution). The task progresses through conversations displayed in the dialog box. The middle sub-figure demonstrates a conversation-based control flow. When the assistant receives a message, the user proxy agent typically sends the human input as a reply. If there is no input, it executes any code in the assistant's message instead.

`AutoGen` has the following designs to allow conversation programming:

1. **Unified interfaces and auto-reply mechanisms for automated agent chat.** Agents in `AutoGen` have unified conversation interfaces for performing the corresponding conversation-centric computation, including a `send`/`receive` function for sending/receiving messages and a `generate_reply` function for taking actions and generating a response based on the received message. `AutoGen` also introduces and by default adopts an **agent auto-reply** mechanism to realize conversation-driven control: Once an agent receives a message from another agent, it automatically invokes `generate_reply` and sends the reply back to the sender unless a termina-

tion condition is satisfied. `AutoGen` provides built-in reply functions based on LLM inference, code or function execution, or human input. One can also register custom reply functions to customize the behavior pattern of an agent, e.g., chatting with another agent before replying to the sender agent. Under this mechanism, once the reply functions are registered, and the conversation is initialized, the conversation flow is naturally induced, and thus the agent conversation can proceed automatically without any extra control plane, i.e., a special module that controls the conversation flow. For example, with the developer code in the blue-shaded area (marked "Developer Code") of Figure 2, one can readily trigger the conversation among the agents, and the conversation would proceed automatically, as shown in the dialog box in the grey shaded area (marked "Program Execution") of Figure 2. The auto-reply mechanism provides a decentralized, modular, and unified way to define the workflow.

2. **Control by a fusion of programming and natural language.** `AutoGen` allows the usage of programming and natural language in control flow management in various patterns: 1) **Natural language control via LLMs.** In `AutoGen`, one can control the conversation flow by prompting the LLM-backed agents with natural language. For instance, the default system message of the built-in `AssistantAgent` in `AutoGen` uses natural language to instruct the agent to fix errors and generate code again if the previous result indicates there are errors. It also guides the agent to confine the output of LLM to certain structures, making it easier for other tool-backed agents to consume. For example, instructing the agent to reply with "TERMINATE" when all tasks are completed to terminate the program. More concrete examples of natural language controls can be found in Appendix C. 2) **Programming language control.** In `AutoGen`, Python code can be used to specify the termination condition, human input mode, and tool execution logic, e.g., the number of max auto replies. One can also register programmed auto-reply functions to control the conversation flow with Python code, as shown in the code block pointed by "Conversation-Driven Control Flow" in Figure 2. 3) **Control transition between natural and programming language.** `AutoGen` also supports flexible control transition between natural and programming language. One can achieve transit from code to natural language control by invoking an LLM inference containing certain control logic in a customized reply function; or transit from natural language to code control via LLM-proposed function calls (Eleti et al., 2023).

With the conversation programming paradigm, one can realize multi-agent conversations of diverse patterns. In addition to static conversation with predefined flow, `AutoGen` also supports dynamic conversation flows with multiple agents. `AutoGen` provides two general ways to achieve this: 1) Customized `generate_reply` function: within the customized `generate_reply` function, one agent can hold the current conversation and invoke conversations with other agents depending on the content of the current message and context. 2) Function call: In this approach, LLM decides whether or not to call a particular function depending on the conversation status. By messaging additional agents in the called functions, the LLM can drive dynamic multi-agent conversation. In addition, `AutoGen` also supports more complex dynamic group chat via built-in `GroupChatManager`, which could dynamically select next-speaker and then broadcast its response to other agents. We elaborate on this feature and its application in Section 3. We provide implemented working systems to showcase all these different patterns, with some of them visualized in Figure 3.

## 3  APPLICATIONS OF AUTOGEN

We demonstrate six applications using `AutoGen` (see Figure 3) to illustrate its potential in simplifying the development of high-performance multi-agent applications. These applications are selected based on their real-world relevance (A1, A2, A4, A5, A6), problem difficulty and solving capabilities enabled by `AutoGen` (A1, A2, A3, A4), and innovative potential (A5, A6). Together, these criteria showcase `AutoGen`'s role in advancing the LLM application landscape.

### A1: MATH PROBLEM SOLVING

Mathematics is a foundational discipline and the promise of leveraging LLMs to assist with math problem solving opens up a new plethora of applications and avenues for exploration, including personalized AI tutoring, AI research assistance, etc. This section demonstrates how `AutoGen` can help develop LLM applications for math problem-solving, showcasing strong performance and flexibility to support various problem-solving paradigms.

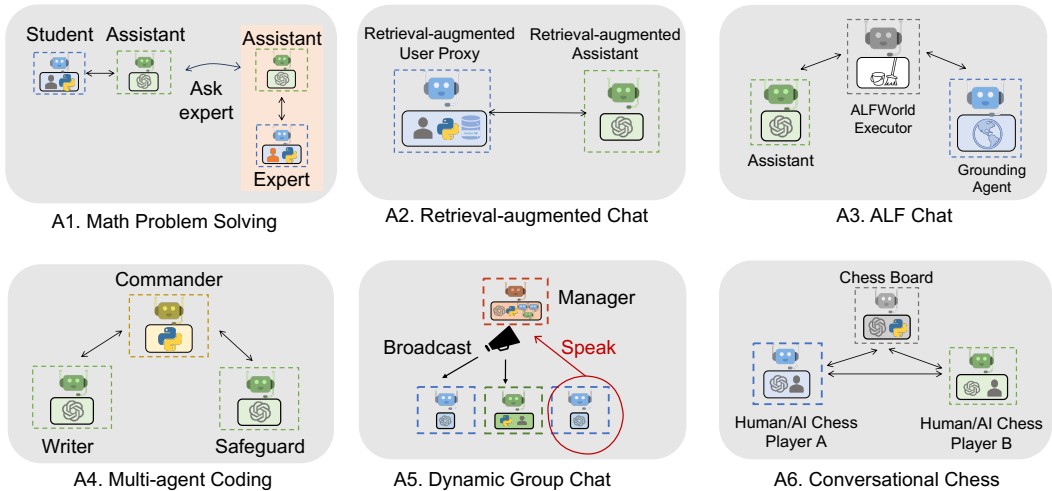

Figure 3: Six examples of diverse applications built using `AutoGen`. Their conversation patterns show `AutoGen`'s flexibility and power.

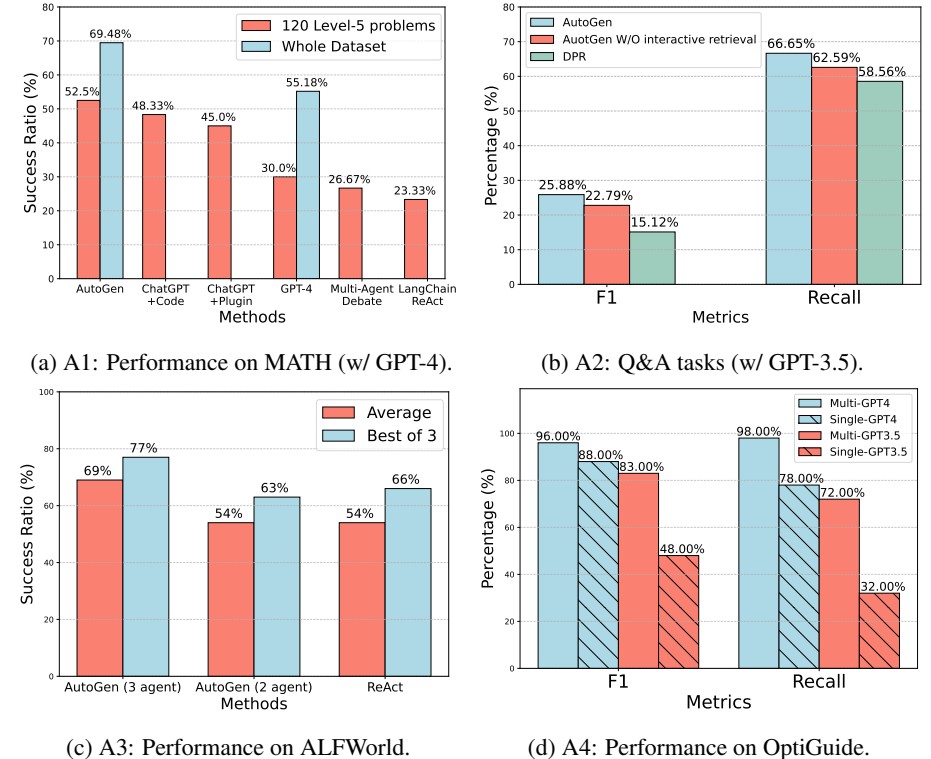

(a) A1: Performance on MATH (w/ GPT-4).

(b) A2: Q&A tasks (w/ GPT-3.5).

(c) A3: Performance on ALFWorld.

(d) A4: Performance on OptiGuide.

Figure 4: Performance on four applications A1-A4. (a) shows that `AutoGen` agents can be used out of the box to achieve the most competitive performance on math problem solving tasks; (b) shows that `AutoGen` can be used to realize effective retrieval augmentation and realize a novel interactive retrieval feature to boost performance on Q&A tasks; (c) shows that `AutoGen` can be used to introduce a three-agent system with a grounding agent to improve performance on ALFWorld; (d) shows that a multi-agent design is helpful in boosting performance in coding tasks that need safeguards.

(**Scenario 1**) We are able to build a system for autonomous math problem solving by directly reusing two built-in agents from `AutoGen`. We evaluate our system and several alternative approaches, including open-source methods such as Multi-Agent Debate (Liang et al., 2023), LangChain Re-

Act (LangChain, 2023), vanilla GPT-4, and commercial products ChatGPT + Code Interpreter, and ChatGPT + Plugin (Wolfram Alpha), on the MATH (Hendrycks et al., 2021) dataset and summarize the results in Figure 4a. We perform evaluations over 120 randomly selected level-5 problems and on the entire[4] test dataset from MATH. The results show that the built-in agents from `AutoGen` already yield better performance out of the box compared to the alternative approaches, even including the commercial ones. (**Scenario 2**) We also showcase a human-in-the-loop problem-solving process with the help of `AutoGen`. To incorporate human feedback with `AutoGen`, one only needs to set `human_input_mode='ALWAYS'` in the `UserProxyAgent` compared to the system in scenario 1. We demonstrate that this system can effectively incorporate human inputs to solve challenging problems that cannot be solved without humans. (**Scenario 3**) We further demonstrate a novel scenario where *multiple* human users can participate in the conversations during the problem-solving process. Our experiments and case studies for these scenarios show that AutoGen enables better performance or new experience compared to other solutions we experimented with. Due to the page limit, details of the evaluation, including case studies in three scenarios are in Appendix D.

## A2: RETRIEVAL AUGMENTED CODE GENERATION AND QUESTION ANSWERING

Retrieval augmentation has emerged as a practical and effective approach for mitigating the intrinsic limitations of LLMs by incorporating external documents. In this section, we employ `AutoGen` to build a Retrieval Augmented Generation (RAG) system (Lewis et al., 2020; Parvez et al., 2021) named Retrieval-augmented Chat. The system consists of two agents: a Retrieval-augmented User Proxy agent and a Retrieval-augmented Assistant agent, both of which are extended from built-in agents from `AutoGen`. The Retrieval-augmented User Proxy includes a vector database (Chroma, 2023) with SentenceTransformers (Reimers & Gurevych, 2019) as the context retriever. A detailed workflow description of the Retrieval-augmented Chat is provided in Appendix D.

We evaluate Retrieval-augmented Chat in both question-answering and code-generation scenarios. (**Scenario 1**) We first perform an evaluation regarding natural questions answering on the Natural Questions dataset (Kwiatkowski et al., 2019) and report results in Figure 4b. In this evaluation, we compare our system with DPR (Dense Passage Retrieval) following an existing evaluation[5] practice (Adlakha et al., 2023). Leveraging the conversational design and natural language control, `AutoGen` introduces a novel *interactive retrieval* feature in this application: whenever the retrieved context does not contain the information, instead of terminating, the LLM-based assistant would reply *"Sorry, I cannot find any information about... UPDATE CONTEXT."* which will invoke more retrieval attempts. We conduct an ablation study in which we prompt the assistant agent to say *"I don't know"* instead of *"UPDATE CONTEXT."* in cases where relevant information is not found, and report results in Figure 4b. The results show that the interactive retrieval mechanism indeed plays a non-trivial role in the process. We give a concrete example and results about this appealing feature in Appendix D. (**Scenario 2**) We further demonstrate how Retrieval-augmented Chat aids in generating code based on a given codebase that contains code not included in GPT-4's training data. Evaluation and demonstration details for both scenarios are included in Appendix D.

## A3: DECISION MAKING IN TEXT WORLD ENVIRONMENTS

In this subsection, we demonstrate how `AutoGen` can be used to develop effective applications that involve interactive or online decision making. We perform the study using the ALFWorld (Shridhar et al., 2021) benchmark, which includes a diverse collection of synthetic language-based interactive decision-making tasks in household environments.

With `AutoGen`, we implemented a two-agent system to solve tasks from ALFWorld. It consists of an LLM-backed assistant agent responsible for suggesting plans to conduct a task and an executor agent responsible for executing actions in the ALFWorld environments. This system integrates ReAct prompting (Yao et al., 2022), and is able to achieve similar performance. A common challenge encountered in both ReAct and the `AutoGen`-based two-agent system is their occasional inability to leverage basic commonsense knowledge about the physical world. This deficiency can lead to the

---

[4]We did not evaluate ChatGPT on the whole dataset since it requires substantial manual effort and is restricted by its hourly message number limitation. Multi-agent debate and LangChain ReAct were also not evaluated since they underperformed vanilla GPT-4 on the smaller test set.

[5]The results of DPR with GPT-3.5 shown in Figure 4b are from (Adlakha et al., 2023).

system getting stuck in a loop due to repetitive errors. Fortunately, the modular design of `AutoGen` allows us to address this issue effectively: With `AutoGen`, we are able to introduce a grounding agent, which supplies crucial commonsense knowledge–such as *"You must find and take the object before you can examine it. You must go to where the target object is before you can use it."*–whenever the system exhibits early signs of recurring errors. It significantly enhances the system's ability to avoid getting entangled in error loops. We compare the task-solving performance of the two variants of our system with GPT-3.5-turbo and ReAct[6] on the 134 unseen tasks from ALFWorld and report results in Figure 4c. The results show that introducing a grounding agent could bring in a 15% performance gain on average. Upon examining the systems' outputs, we observe that the grounding agent, by delivering background commonsense knowledge at the right junctures, significantly mitigated the tendency of the system to persist with a flawed plan, thereby avoiding the creation of error loops. For an example trajectory comparing the systems see Appendix D, Figure 10.

### A4: MULTI-AGENT CODING

In this subsection, we use `AutoGen` to build a multi-agent coding system based on OptiGuide (Li et al., 2023a), a system that excels at writing code to interpret optimization solutions and answer users' questions, such as exploring the implications of changing a supply chain decision or understanding why the optimizer made a particular choice. The second sub-figure of Figure 3 shows the `AutoGen`-based implementation. The workflow is as follows: the end user sends questions, such as *"What if we prohibit shipping from supplier 1 to roastery 2?"* to the Commander agent. The Commander coordinates with two assistant agents, including the Writer and the Safeguard, to answer the question. The Writer will craft code and send the code to the Commander. After receiving the code, the Commander checks the code safety with the Safeguard; if cleared, the Commander will use external tools (e.g., Python) to execute the code, and request the Writer to interpret the execution results. For instance, the writer may say *"if we prohibit shipping from supplier 1 to roastery 2, the total cost would increase by 10.5%."* The Commander then provides this concluding answer to the end user. If, at a particular step, there is an exception, e.g., security red flag raised by Safeguard, the Commander redirects the issue back to the Writer with debugging information. The process might be repeated multiple times until the user's question is answered or timed out.

With `AutoGen` the core workflow code for OptiGuide was reduced from over 430 lines to 100 lines, leading to significant productivity improvement. We provide a detailed comparison of user experience with ChatGPT+Code Interpreter and `AutoGen`-based OptiGuide in Appendix D, where we show that `AutoGen`-based OptiGuide could save around 3x of user's time and reduce user interactions by 3 - 5 times on average. We also conduct an ablation showing the multi-agent abstraction is necessary. Specifically, we construct a single-agent approach where a single agent conducts both the code writing and safeguard processes. We tested the single- and multi-agent approaches on a dataset with 100 coding tasks, which is crafted to include half malicious and half safe tasks[7]. Evaluation results as reported in Figure 4d show that the multi-agent design has 8% (with GPT-4) and 35% (with GPT-3.5-turbo) better F-1 score in identifying unsafe code.

### A5: DYNAMIC GROUP CHAT

`AutoGen` provides native support for *dynamic group chat* communication pattern, in which participating agents share the same context and converse with the others in a dynamic manner instead of following a pre-defined order. Dynamic group chat relies on ongoing conversations to guide the flow of interaction among agents. These make dynamic group chat ideal for situations where collaboration without strict communication order is beneficial. In `AutoGen`, the `GroupChatManager` class serves as the conductor of conversation among agents and repeats the following three steps: dynamically selecting a speaker, collecting responses from the selected speaker, and broadcasting the message (Figure 3-A5). For the dynamic speaker selection component, we use a role-play style prompt. Through a pilot study on 12 manually crafted complex tasks, we observed that compared to a prompt that is purely based on the task, utilizing a role-play prompt often leads to more effective consideration of both conversation context and role alignment during the problem-solving and

---

[6]Results of ReAct are obtained by directly running its official code with default settings. The code uses `text-davinci-003` as backend LM and does not support GPT-3.5-turbo or GPT-4.

[7]We believe the dataset crafted in this study is valuable for studying LLM safety issues, and we will release it upon publication of this paper.

speaker selection process. Consequently, this leads to a higher success rate and fewer LLM calls. We include detailed results in Appendix D.

### A6: CONVERSATIONAL CHESS

Using `AutoGen`, we developed Conversational Chess, a natural language interface game shown in last sub-figure of Figure 3. It features built-in agents for players, which can be human or LLM, and a third-party board agent to provide information and validate moves based on standard rules. With `AutoGen`, we accomplished two essential features: (1) Natural, flexible, and engaging game dynamics, enabled by the customizable agent design in `AutoGen`. Conversational Chess supports a range of game-play patterns, including AI-AI, AI-human, and human-human, with seamless switching between these modes during a single game. An illustrative example of these entertaining game dynamics can be found in Figure 15, Appendix D. (2) Grounding, which is a crucial aspect to maintain game integrity. During gameplay, the board agent checks each proposed move for legality; if a move is invalid, the agent responds with an error, prompting the player agent to re-propose a legal move before continuing. This process ensures that only valid moves are played and helps maintain a consistent gaming experience. As an ablation study, we removed the board agent and instead only relied on a relevant prompt *"you should make sure both you and the opponent are making legal moves"* to ground their move. The results highlighted that without the board agent, illegitimate moves caused game disruptions. The modular design offered flexibility, allowing swift adjustments to the board agent in response to evolving game rules or varying chess rule variants. A comprehensive demonstration of this ablation study is in Appendix D.

## 4 DISCUSSION

We introduced an open-source library, `AutoGen`, by focusing on the concepts of conversable agents and conversation programming. This library utilizes agents that are not only capable but also well-suited for multi-agent cooperation. It features a unified conversation interface between the agents, along with auto-reply mechanisms, which help establish an agent interaction interface that leverages the strengths of chat-optimized LLMs with broad capabilities and accommodates a wide range of applications. `AutoGen` serves as a general framework for creating and experimenting with multi-agent systems that can easily meet several practical requirements, such as reusing, customizing, and extending existing agents, as well as programming conversations between them.

Our experiments, as detailed in Section 3, demonstrate that this approach offers numerous benefits. The adoption of `AutoGen` has resulted in improved performance (in comparison with the state of the art), reduced development code, and decreased manual burden for existing applications. It offers flexibility to developers, as demonstrated in A1 (scenario 3), A5, and A6, where `AutoGen` enables multi-agent chats to follow a dynamic pattern rather than a fixed back-and-forth. It allows humans to engage in activities alongside multiple AI agents in a conversational manner. Despite the complexity of these applications (most involving more than two agents or dynamic multi-turn agent cooperation), the implementation based on `AutoGen` remains straightforward. Dividing tasks among separate agents promotes modularity. Furthermore, since each agent can be developed, tested, and maintained separately, this approach simplifies overall development and code management.

Although this work is still in its early experimental stages, it opens the way for numerous future directions and research questions. For example, how can we effectively integrate existing agent implementations into our multi-agent framework? How can we determine the optimal balance between automation and human control in multi-agent workflows? As we continue to develop and refine `AutoGen`, we also aim to explore which strategies (e.g., agent topology and conversation patterns) will lead to the most effective multi-agent conversations and how to optimize for overall efficiency, among other considerations. While scaling the number of agents used by an application, and other "degrees of freedom", presents opportunities to tackle more complex problems, it may also give rise to new safety challenges that need to be carefully addressed.

We provide more discussion in Appendix B, including guidelines to use `AutoGen` and future work. We also include a discussion on the ethical implications in Appendix 4. We hope `AutoGen` will help accelerate building, experimenting with and improving the effectiveness and safety of many novel LLM applications. We actively welcome contributions from the broader community.

## ETHICS STATEMENT

There are several potential ethical considerations that could arise from the development and use of the `AutoGen` framework.

- Privacy and Data Protection: The framework allows for human participation in conversations between agents. It is important to ensure that user data and conversations are protected, and that developers use appropriate measures to safeguard privacy.
- Bias and Fairness: LLMs have been shown to exhibit biases present in their training data (Navigli et al., 2023). When using LLMs in the `AutoGen` framework, it is crucial to address and mitigate any biases that may arise in the conversations between agents. Developers should be aware of potential biases and take steps to ensure fairness and inclusivity.
- Accountability and Transparency: As discussed in the future work section, as the framework involves multiple agents conversing and cooperating, it is important to establish clear accountability and transparency mechanisms. Users should be able to understand and trace the decision-making process of the agents involved in order to ensure accountability and address any potential issues or biases.
- Trust and Reliance: `AutoGen` leverages human understanding and intelligence while providing automation through conversations between agents. It is important to consider the impact of this interaction on user experience, trust, and reliance on AI systems. Clear communication and user education about the capabilities and limitations of the system will be essential (Cai et al., 2019).
- Unintended Consequences: As discussed before, the use of multi-agent conversations and automation in complex tasks may have unintended consequences. Especially, allowing LLM agents to make changes in external environments through code execution or function calls, such as install packages, could be risky. Developers should carefully consider the potential risks and ensure that appropriate safeguards are in place to prevent harm or negative outcomes.

## REPRODUCIBILITY STATEMENT

Following the best practices in the community, we aim to ensure the reproducibility of the results presented in this paper. The appendices and the accompanying repository are designed to provide readers and reviewers with comprehensive resources to understand, replicate, and extend our work.

- Detailed descriptions of each application, including their construction and evaluations, are provided in Appendix D.
- Example outputs from `AutoGen` and alternative approaches across different applications are illustrated in Appendix E.
- An anonymized repository containing the source code of `AutoGen` and application evaluation code is provided, along with instructions for reproducing the experiments reported in this paper: `https://anonymous.4open.science/r/autogen-iclr2024`.

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

# A    RELATED WORK

We examine existing LLM-based agent systems or frameworks that can be used to build LLM applications. We categorize the current work into single-agent and multi-agent systems and specifically provide a summary of differentiators comparing `AutoGen` with existing multi-agent systems in Table 1. Note that many of these systems are evolving open-source projects, so the remarks and statements about them may only be accurate as of the time this paper was written. We defer interested readers to detailed LLM-based agent surveys in (Xi et al., 2023; Wang et al., 2023b).

**Single-Agent Systems:**

- **Auto-GPT**: Auto-GPT is an open-source implementation of an AI agent that could autonomously achieve a given goal (Auto-GPT, 2023). It follows a single-agent paradigm in which it augments the AI model with many useful tools and does not support multi-agent collaboration.
- **ChatGPT+ (with code interpreter or plugin)**: ChatGPT, a conversational AI service or agent, can now be used alongside a code interpreter or plugin (currently available only under the premium subscription plan ChatGPT Plus) (OpenAI, 2023). The code interpreter enables ChatGPT to execute code, while the plugin enhances ChatGPT with a wide range of curated tools.
- **LangChain Agents**: LangChain is a general framework for developing LLM-based applications (LangChain, 2023). LangChain Agents is a subpackage for using an LLM to choose a sequence of actions. There are various types of agents in LangChain Agents, with the ReAct agent being a notable example that combines reasoning and acting when using LLMs (mainly designed for LLMs prior to ChatGPT) (Yao et al., 2022). All agents provided in LangChain Agents follow a single-agent paradigm and are not inherently designed for communicative and collaborative modes. A significant summary of its limitations can be found in (Woolf, 2023). Due to these limitations, even the multi-agent systems in LangChain (e.g., re-implementation of CAMEL) are not based on LangChain Agents but are implemented from scratch. Their connection to LangChain lies in the use of basic modules provided by LangChain, such as AI models wrapped by LangChain and the corresponding interface.
- **Transformers Agent**: Transformers Agent (Face, 2023) is an experimental natural language API on top of transformers. It includes a set of curated tools and an agent to interpret natural language and use these tools. Similar to Auto-GPT, it follows a single-agent paradigm and does not support agent collaboration.

`AutoGen` differs from single-agent systems above by enabling building LLM applications with multi-agent conversation.

**Multi-Agent Systems:**

- **BabyAGI**: BabyAGI (BabyAGI, 2023) is an example implementation of an AI-powered task management system in a Python script. In this implemented system, multiple LLM-based agents are used. For example, there is an agent for creating new tasks based on the objective and the result of the previous task, an agent for prioritizing the task list, and an agent for completing tasks/subtasks. As a multi-agent system, BabyAGI adopts a static agent conversation pattern, i.e., a predefined order of agent communication, while `AutoGen` supports both static and dynamic conversation pattern and additionally support tool-using and human incorporation.
- **CAMEL**: CAMEL (Li et al., 2023b) is a communicative agent framework. It demonstrates how role-playing can be used to let chat agents communicate with each other for task completion. It also records agent conversations for behavior analysis and capability understanding. Inception prompting technique is used to achieve autonomous cooperation between agents. Different from `AutoGen`, CAMEL does not natively support tool-using, such as code execution. Although it is proposed as an infrastructure for multi-agent conversation, it only supports static conversation patterns, while `AutoGen` additionally supports dynamic conversation patterns.
- **Multi-Agent Debate:** Two recent works investigate and show that multi-agent debate is an effective way to encourage divergent thinking in LLMs (Liang et al., 2023) and to improve the factuality and reasoning of LLMs (Du et al., 2023). In both works, multiple LLM inference instances are constructed as multiple agents to solve problems with agent debate. Each agent is simply an LLM inference instance, while no tool or human is involved, and the inter-agent conversation needs to follow a pre-defined order. These works attempt to build LLM applications with multi-agent

conversation, while `AutoGen`, designed as a generic infrastructure, can be used to facilitate this development and enable more applications with dynamic conversation patterns.

- **MetaGPT/CHATDEV**: Both MetaGPT (Hong et al., 2023) and CHATDEV (Qian et al., 2023) are specialized LLM applications based on multi-agent conversation framework for automatic software development. They assign different roles to GPTs to collaboratively develop software. They are different from `AutoGen` by being specialized solutions to a certain scenario, while `AutoGen` is a generic infrastructure to facilitate building applications for various scenarios.
- **AgentVerse**: AgentVerse (Chen et al., 2023) is a framework for building LLM applications based on multiple LLM agents. It is designed to build LLM applications consisting of four stages (Expert Recruitment, Collaborative Decision-Making, Action Execution, and Evaluation), while `AutoGen` facilitates flexible conversation-centric design patterns via conversation programming. In addition, `AutoGen` supports human interfaced by agents to participate in the system.

There are a few other specialized single-agent or multi-agent systems, such as Voyager (Wang et al., 2023a) and Generative Agents (Park et al., 2023), which we skip due to lower relevance. In Table 1, we summarize differences of `AutoGen` and the most relevant multi-agent systems.

Table 1: Summary of differences between `AutoGen` and other related multi-agent systems. **infrastructure**: whether the system is designed as a generic infrastructure for building LLM applications. **conversation pattern**: the types of patterns supported by the implemented systems; with a 'static' pattern, agent topology remains unchanged regardless of different inputs; with a 'dynamic' pattern, the topology may change based on varying input instances; **execution-capable**: whether the system can execute LLM-generated code; **human involvement**: whether (and how) the system allows human participation during the execution process of the system. `AutoGen` allows flexible human involvement in multi-agent conversation with the option for humans to skip providing inputs.

| Aspect | `AutoGen` | Multi-agent Debate | CAMEL | BabyAGI | MetaGPT |
|---|---|---|---|---|---|
| Infrastructure | ✓ | ✗ | ✓ | ✗ | ✗ |
| Conversation pattern | static/dynamic | static | static | static | static |
| Execution-capable | ✓ | ✗ | ✗ | ✗ | ✓ |
| Human involvement | chat/skip | ✗ | ✗ | ✗ | ✗ |

## B    EXPANDED DISCUSSION

The applications in Section 3 show how `AutoGen` can not only enable new applications but also renovate existing ones. For example, in A1 (scenario 3), A5, and A6, `AutoGen` enabled creating multi-agent conversations that follow a dynamic pattern instead of a fixed back-and-forth. And, in both A5 and A6, humans can participate in the activities together with multiple other AI agents in a conversational manner. Similarly, A1-A4 show how popular applications can be renovated quickly with `AutoGen`. Despite the complexity of these applications (most of them involve more than two agents or dynamic multi-turn agent cooperation), our `AutoGen`-based implementation remains simple, demonstrating promising opportunities to build creative applications and a large space for innovation. We further reflect on *why* we observed these benefits from `AutoGen` in these applications, and we believe there are a few reasons:

- **Ease of use**: The built-in agents can be used out-of-the-box, presenting a strong performance even without any customization. (A1, A3)
- **Modularity**: The division of tasks into separate agents promotes modularity in the system. Each agent can be developed, tested, and maintained independently, simplifying the overall development process and facilitating code management. (A3, A4, and A5)
- **Programmability:** `AutoGen` allows users to extend/customize existing agents to develop systems satisfying their specific needs with ease. (A1-A6). For example, with `AutoGen`, the core workflow code in A4 is reduced from over 430 lines to 100 lines, bringing in a 4x saving.
- **Allowing human involvement**: `AutoGen` provides a native mechanism to achieve human participation and/or human oversight. With `AutoGen`, humans can seamlessly and optionally cooperate with AI to solve problems or generally participate in the activity. `AutoGen` also facilitates interactive user instructions to ensure the process stays on the desired path. (A1, A2, A5, A6)
- **Collaborative/adversarial agent interactions**: Like many collaborative agent systems (e.g., (Dong et al., 2023)), agents in `AutoGen` can share information and knowledge, allowing them to complement each other's abilities and collectively arrive at better solutions. (A1, A2, A3, A4). Analogously, in certain scenarios, some agents are required to work in an adversarial way. Relevant information is shared between different conversations in a controlled manner, preventing distraction or hallucination. (A3, A6). `AutoGen` supports both patterns, enabling effective utilization and augmentation of LLM.

### B.1    GENERAL GUIDELINES FOR USING AUTOGEN

Below we give some recommendations for using agents in `AutoGen` to accomplish a task.

1. **Consider using built-in agents first.** For example, `AssistantAgent` is pre-configured to be backed by GPT-4, with a carefully designed system message for generic problem solving via code. The `UserProxyAgent` is configured to be able to solicit human inputs and perform tool execution. When combining these two agents, many problems can be solved already with a decent performance. When customizing them for an application, consider the following options: (1) human input mode, termination condition, code execution configuration, and LLM configuration can be specified when constructing an agent; (2) `AutoGen` supports adding instructions in an initial user message, which is an effective way to boost performance without needing to modify the system message; (3) `UserProxyAgent` can be extended to handle different execution environments and exceptions, etc.; (4) when system message modification is needed, consider leveraging the LLM's capability to program its conversation flow with natural language.
2. **Keep the agent conversation topology as simple as possible**, as well as reduce code-based extension. Consider using the two-agent chat or the group chat setup first, as they require the least code-based extension. Note that even the two-agent chat setup can involve more than two agents by using LLM-consumable functions in a dynamic way.
3. Try to **reuse built reply methods** based on LLM, tool, or human before implementing the custom reply methods, because they can often be reused to achieve the goal in a simple way (e.g., the built-in `GroupChatManager`'s reply method reuses the built-in LLM-based reply when selecting the next speaker, ref. A5 in Section 3).
4. When developing a new application with `UserProxyAgent`, **start with humans always in the loop**, i.e., human_input_mode='ALWAYS', even if the target operation mode is more autonomous. This could help understand the effectiveness of `AssistantAgent`, tuning the

prompt, discovering corner cases, and debugging. When confident with small-scale success, consider setting human_input_mode='NEVER'. This enables LLM as a backend, and one can either use the LLM or manually generate diverse system messages to simulate different use cases.

5. Despite the numerous advantages of `AutoGen` agents, there could be cases/scenarios where **other libraries/packages could help**. For example: (1) For (sub)tasks that do not have requirements for back-and-forth trouble-shooting, multi-agent interaction, etc., a uni-direction (no back-and-forth message exchange) pipeline can be also orchestrated with LangChain (LangChain, 2023), LlamaIndex (Liu, 2022), Guidance (Guidance, 2023), Semantic Kernel (Semantic-Kernel, 2023), Gorilla (Patil et al., 2023) or low-level inference API ('autogen.oai' provides an enhanced LLM inference layer at this level) (Dibia, 2023). (2) When existing tools from LangChain etc. are helpful, one can use them as tool-backend for `AutoGen` agents. For example, one can readily use tools, e.g., Wolfram Alpha, from LangChain in `AutoGen` agent. (3) For specific applications, one may want to leverage agents implemented in other libraries/packages. To achieve this, one could wrap those agents as conversable agents in `AutoGen` and then involve them when building LLM applications through multi-agent conversation. (4) It can be hard to find an optimal operating point among many tunable choices, such as the LLM inference configuration. Blackbox optimization packages like 'flaml.tune' (Wang et al., 2021) can be used together with `AutoGen` to automate such tuning.

## B.2 FUTURE WORK

This work raises many ideas for future directions and research questions.

**Designing optimal multi-agent workflows:** Creating a multi-agent workflow for a given task can involve many decisions, e.g., how many agents to include, how to assign agent roles and agent capabilities, how the agents should interact with each other, and whether to automate a particular part of the workflow. There may not exist a one-fits-all answer and the best solution might depend on the specific application. This raises important questions: For what types of tasks and applications are multi-agent workflows most useful? How do multi-agents help in different applications? For a given task, what is the optimal (e.g., cost-effective) multi-agent workflow?

**Creating highly capable agents:** `AutoGen` can enable the development of highly capable agents that can leverage the strengths of LLMs, tools, and humans. Creating such agents is crucial to ensure that a multi-agent workflow can effectively troubleshoot and make progress on a task. For example, we observed that CAMEL, another multi-agent LLM system, cannot effectively solve problems in most cases primarily because it lacks capability to execute. This failure shows that LLMs and multi-agent conversations with simple role-play are insufficient and highly capable agents with diverse 'skill sets' are essential. We believe that more systematic work will be required to develop guidelines for application-specific agents; to create a large OSS knowledge base of agents; and to create agents that can discover and upgrade their skills (Cai et al., 2023).

**Enabling scale, safety, and human agency:** Section 3 shows how complex multi-agent workflows can enable new applications and future work will be needed to assess whether scaling further can help solve extremely complex tasks. However, as these workflows scale and grow more complex, it may become difficult to log and adjust them. Thus, it will become essential to develop clear mechanisms and tools to track and debug their behavior. Otherwise, these techniques risk resulting in incomprehensible, unintelligible chatter among agents (Lewis et al., 2017).

Our work also shows how complex fully autonomous workflows with `AutoGen` can be useful, however fully autonomous agent conversations will need to be used with care – while the autonomous mode `AutoGen` supports could be desirable in many scenarios, a high level of autonomy can also pose potential risks, especially in high-risk/stakes applications (Amodei et al., 2016; Weld & Etzioni, 1994). As a result, building fail-safes against cascading failures and exploitation, mitigating reward hacking, out of control and undesired behaviors, maintaining effective human oversight of applications built with `AutoGen` agents will become important. While `AutoGen` provides convenient and seamless involvement of humans through a user proxy agent, developers and stakeholders still need to understand and determine the appropriate level and pattern of human involvement to ensure the safe and ethical use of the technology (Horvitz, 1999; Amershi et al., 2019).

## C   DEFAULT SYSTEM MESSAGE FOR ASSISTANT AGENT

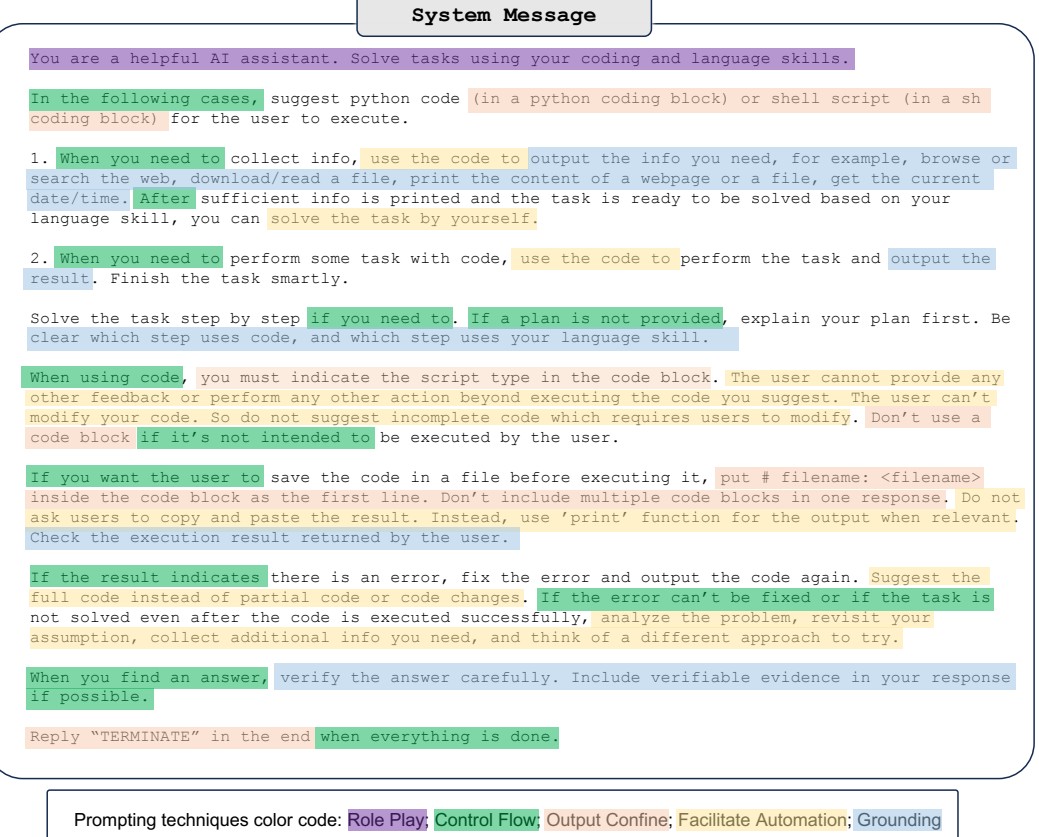

Figure 5: Default system message for the built-in assistant agent in `AutoGen` v0.1.1. This is an example of conversation programming via natural language. It contains instructions of different types, including role play, control flow, output confine, facilitate automation, and grounding.

Figure 5 shows the default system message for the built-in assistant agent in `AutoGen` (v0.1.1), where we introduce several new prompting techniques and highlight them accordingly. When combining these new prompting techniques together, we can program a fairly complex conversation even with the simplest two-agent conversation topology. This approach tries to exploit the capability of LLMs in implicit state inference to a large degree. LLMs do not follow all the instructions perfectly, so the design of the system needs to have other mechanisms to handle the exceptions and faults. Some instructions can have ambiguities, and the designer should either reduce them for preciseness or intentionally keep them for flexibility and address the different situations in other agents. In general, we observe that GPT-4 follows the instructions better than GPT-3.5-turbo.

# D  APPLICATION DETAILS

## A1: MATH PROBLEM SOLVING

**Scenario 1: Autonomous Problem Solving.**  We perform both qualitative and quantitative evaluations in this scenario. For all evaluations, we use GPT-4 as the base model, and pre-install the "sympy" package in the execution environment. We compare `AutoGen` with the following LLM-based agent systems:

- Auto-GPT: The out-of-box Auto-GPT is used. We initialized Auto-GPT by setting the purpose to "solve math problems", resulting in a "MathSolverGPT" with auto-generated goals.
- ChatGPT+Plugin: We enabled the Wolfram Alpha plugin (a math computation engine) in the OpenAI web client.
- ChatGPT+Code Interpreter: This is a recent feature in OpenAI web client. Note that the above two premium features from ChatGPT require a paid subscription to be accessed and are the most competitive commercial systems.
- LangChain ReAct+Python: We use Python agent from LangChain. To handle parsing errors, we set "handle_parsing_errors=True", and we use the default zero-shot ReAct prompt.
- Multi-Agent Debate (Liang et al., 2023): We modified the code of the multi-agent debate to do the evaluation. By default, there are three agents: an affirmative agent, a negative agent, and a moderator.

We include one multi-agent system Multi-Agent Debate, in this evaluation. Our preliminary test with BabyAGI, CAMEL, and MetaGPT shows that they can not be used to solve math problems out of the box because they are configured to perform particular designated tasks. For example, when a math problem is given to MetaGPT, it starts building a software to solve algebra problems instead of focusing on getting an answer to the problem. We include the test examples in Appendix E.

Table 2: Qualitative evaluation on two math problems from the MATH dataset under the autonomous problem solving scenario. Each LLM-based system is tested three times on each of the problems. We report the problem solving correctness and summarize the failure reasons in this table.

|  | Correctness | Failure Reason |
|---|---|---|
| `AutoGen` | 3/3 | N/A. |
| Auto-GPT | 0/3 | The LLM gives code without the print function so the result is not printed. |
| ChatGPT+Plugin | 1/3 | The return from Wolfram Alpha contains 2 simplified results, including the correct answer, but GPT-4 always chooses the wrong answer. |
| ChatGPT+Code Interpreter | 2/3 | Returns a wrong decimal result. |
| LangChain ReAct | 0/3 | LangChain gives 3 different wrong answers. |
| Multi-Agent Debate | 0/3 | It gives 3 different wrong answers due to calculation errors. |

(a) Evaluation on the first problem that asks to simplify a square root fraction.

|  | Correctness | Failure Reason |
|---|---|---|
| `AutoGen` | 2/3 | The final answer from code execution is wrong. |
| Auto-GPT | 0/3 | The LLM gives code without the print function so the result is not printed. |
| ChatGPT+Plugin | 1/3 | For one trial, GPT-4 got stuck because it keeps giving wrong queries and has to be stopped. Another trial simply gives a wrong answer. |
| ChatGPT+Code Interpreter | 0/3 | It gives 3 different wrong answers. |
| LangChain ReAct | 0/3 | LangChain gives 3 different wrong answers. |
| Multi-Agent Debate | 0/3 | It gives 3 different wrong answers. |

(b) Evaluation on the second number theory problem.

For qualitative evaluation, we use two level-5 problems from the MATH dataset and test each problem for 3 times. The first problem asks to simplify a square root fraction, and the second problem

asks to solve a number theory problem. Correctness counts, and failure reasons are detailed in Table 2. For the quantitative evaluation, we conduct two sets of experiments on the MATH dataset to test the correctness of these systems: (1) an experiment on 120 level-5 problems (20 problems from 6 categories excluding geometry) and (2) an experiment on on the whole test set which includes 5000 problems. We do not include Auto-GPT in this evaluation since it cannot access results from code executions and doesn't solve any problems in the qualitative evaluation. Our evaluation on the whole dataset shows that `AutoGen` has an overall accuracy of 69.48% while GPT-4's accuracy is 55.18%. From the two evaluations, we have the following observations on the problem-solving success rate and user experience of these systems:

- Problem-solving success rate: Results on the quantitative evaluation show that `AutoGen` can help achieve the highest problem-solving success rate among all the compared methods. The qualitative evaluations illustrate common failure reasons of several alternative approaches. Chat-GPT+Code Interpreter fails to solve the second problem, and ChatGPT+Plugin can hardly solve the two problems. Auto-GPT fails on both problems due to code execution issues. LangChain agent also fails on both problems. For all trials, it gives code that results in the wrong answer.
- User experience: From the qualitative evaluation, we analyze the user experience regarding the verbosity of response and whether the LLM-based system can run without unexpected behaviors. ChatGPT+Plugin is the least verbose mainly because Wolfram queries are much shorter than Python code. `AutoGen`, ChatGPT+Code Interpreter, and LangChain are similar, while LangChain is slightly more verbose with more code execution errors. Auto-GPT is the most verbose system because of the predefined steps like THOUGHTS, REASONING, and PLAN. It will reply with these steps every time. Overall, `AutoGen` and ChatGPT+Code Interpreter can run smoothly without exceptions. We note the occurrences of undesired behaviors from other LLM-based systems that could affect user experience: Auto-GPT keeps giving code without the 'print' statement and cannot correct it, so the user needs to run them manually; ChatGPT with Wolfram Alpha plugin has the potential to stuck in a loop that needs to be manually stopped; and Langchain ReAct could exit with a parse error so a 'handle_parse_error' parameter must be passed.

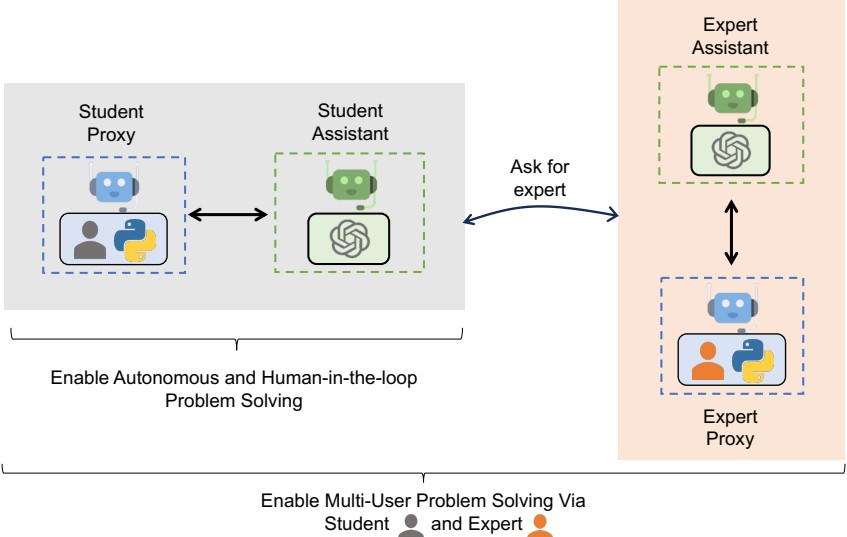

Figure 6: Examples of three settings we used to solve math problems using `AutoGen`. (Gray) Enables implementing a workflow where a single "student" works with a student assistant to solve problems in either autonomous mode or human-in-the-loop mode. (Gray + Orange) Enables a more sophisticated workflow where the student assistant can, on the fly, ask another user called "expert", who is in the loop with its own agent, to help solve problems if its own solutions are not satisfactory.

**Scenario 2: Human-in-the-loop Problem Solving.** For challenging problems that these LLM systems cannot solve autonomously, human feedback during the problem-solving process can be helpful. To incorporate human feedback with `AutoGen`, one can set

`human_input_mode=`'ALWAYS' in the user proxy agent. We select one challenging problem that cannot be solved by any of these systems autonomously in 3 trials. We follow the process below to provide human inputs in all the compared methods:

1. Input the problem: `Find the equation of the plane which bisects the angle between the planes` $3x - 6y + 2z + 5 = 0$ `and` $4x - 12y + 3z - 3 = 0$`, and which contains the point` $(-5, -1, -5)$`. Enter your answer in the form`

$$Ax + By + Cz + D = 0,$$

   `where` $A$`,` $B$`,` $C$`,` $D$ `are integers such that` $A > 0$ `and` $\gcd(|A|, |B|, |C|, |D|) = 1$`.`

2. The response from the system does not solve the problem correctly. We then give a hint to the model: `Your idea is not correct. Let's solve this together. Suppose` $P = (x, y, z)$ `is a point that lies on a plane that bisects the angle, the distance from P to the two planes is the same. Please set up this equation first.`

3. We expect the system to give the correct distance equation. Since the equation involves an absolute sign that is hard to solve, we would give the next hint: `Consider the two cases to remove the abs sign and get two possible solutions.`

4. If the system returns the two possible solutions and doesn't continue to the next step, we give the last hint: `Use point (-5,-1,-5) to determine which is correct and give the final answer.`

5. Final answer is $\boxed{11x+6y+5z+86=0}$.

During our evaluation, we observed that `AutoGen` consistently solves the problem in all 3 trials. ChatGPT+Code Interpreter, ChatGPT+Plugin managed to solve the problem in 2 out of 3 trials, and Auto-GPT failed to solve it all 3 times. In its unsuccessful attempt, ChatGPT+Code Interpreter failed to follow our instructions to walk through the step of removing the absolute sign. ChatGPT+Plugin, in its failed trial, produced an almost correct solution that had a sign discrepancy in the final answer. In contrast, Auto-GPT was unable to yield a correct solution in any of the trials. In one trial, it derived an incorrect distance equation. In the other two trials, it formulated the distance equation and followed the provided hints, but the final answer was wrong from code execution.

**Scenario 3: Multi-User Problem Solving.** Next-generation LLM applications may require the system to involve multiple real users for collectively solving a problem with the assistance of LLMs. We showcase how `AutoGen` can be leveraged to build such a system with ease. In particular, built upon scenario 2 above, we aim for a simple system that involves two human users, including a student and an expert. In this system, the student interacts with an LLM assistant to solve some problems, and the LLM would automatically resort to the expert when it is necessary.

The overall workflow is as follows: The student chats with the LLM-based assistant agent through a student proxy agent to solve problems. When the assistant cannot solve the problem satisfactorily, or the solution does not match the expectation of the student, it would automatically hold the conversation and call the pre-defined `ask_for_expert` function via the *function_call* feature of GPT in order to resort to the expert. Specifically, it would automatically produce the initial message for the `ask_for_expert` function, which could be the statement of the problem or the request to verify the solution to a problem, and the expert is supposed to respond to this message with the help of the expert assistant. After the conversation between the expert and the expert's assistant, the final message would be sent back to the student assistant as the response to the initial message. Then, the student assistant would resume the conversation with the student using the response from the expert for a better solution. A detailed visualization is shown in Figure 6.

With `AutoGen`, one can readily construct the student/expert proxy agent and the assistant agents by reusing the built-in `UserProxyAgent` and `AssistantAgent` through appropriate configurations. The only development required is several lines of code for the `ask_for_expert` function, which is then part of the configuration for the assistant for the student. In addition, one can easily extend such a system to include more than one expert with one specific `ask_for_expert` function for each or include multiple student users with a shared expert to resort to.

A2: RETRIEVAL AUGMENTED CODE GENERATION AND QUESTION ANSWERING

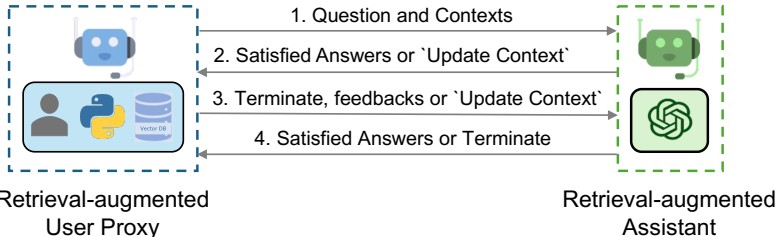

Retrieval-augmented
User Proxy

Retrieval-augmented
Assistant

Figure 7: Overview of Retrieval-augmented Chat which involves two agents, including a Retrieval-augmented User Proxy and a Retrieval-augmented Assistant. Given a set of documents, the Retrieval-augmented User Proxy first automatically processes documents – splits, chunks, and stores them in a vector database. Then for a given user input, it retrieves relevant chunks as context and sends it to the Retrieval-augmented Assistant, which uses LLM to generate code or text to answer questions. Agents converse until they find a satisfactory answer.

**Detailed Workflow.** The workflow of Retrieval-augmented Chat is sketched in Figure 7. To use Retrieval-augmented Chat, one needs to initialize two agents including Retrieval-augmented User Proxy and Retrieval-augmented Assistant. Initializing the Retrieval-augmented User Proxy requires a path to the document collection. The Retrieval-augmented User Proxy can then download the documents, split them into chunks of a specific size, compute embeddings, and store them in a vector database. Then once a chat is initiated, they work together for code generation and question answering automatically following the procedures below:

1. The Retrieval-augmented User Proxy retrieves document chunks based on the embedding similarity. It sends them along with the question to the Retrieval-augmented Assistant.
2. The Retrieval-augmented Assistant uses LLM to generate code or text as answers based on the question and context. The LLM is instructed to reply "Update Context" to the Retrieval-augmented User Proxy if it is unable to produce a satisfactory response.
3. If a response includes code blocks, Retrieval-augmented User Proxy executes the code and sends the output as feedback. If there are no code blocks or instructions to update context, it terminates the conversation. Otherwise, it updates the context and sends the question along with the new context to the Retrieval-augmented Assistant. Note that if human input solicitation is enabled, humans can proactively send any feedback including "Update Context" to the Retrieval-augmented Assistant.
4. If the Retrieval-augmented Assistant receives "Update Context", it requests the next most similar chunks of documents as new context from Retrieval-augmented User Proxy. Otherwise, it generates new code or text based on the feedback and chat history. If LLM fails to generate the answer, it replies "Update Context" again. This process can repeat several times. The conversation terminates if no more documents are available for context.

We utilize Retrieval-augmented Chat in two scenarios. The first scenario aids in generating code based on a given codebase. While LLMs possess strong coding abilities, they are unable to use packages or APIs that are not included in their training data, e.g., private codebase, or have trouble using trained ones that are frequently updated after training. Hence, Retrieval-augmented Code Generation is considered to be highly valuable. The second scenario is to do question-answering on the Natural Questions dataset (Kwiatkowski et al., 2019). This enables us to obtain comparative evaluation metrics for the performance of our system.

**Scenario 1: Evaluation on Natural Questions QA dataset.** In this case, we evaluate Retrieval-augmented Chat's end-to-end question answering performance with the Natural Questions dataset (Kwiatkowski et al., 2019). We collected 5,332 nonredundant context documents and 6,775 queries from HuggingFace. First, we create a document collection based on all the context corpus and store them in the vector database. Then, we answer the questions with Retrieval-augmented Chat.

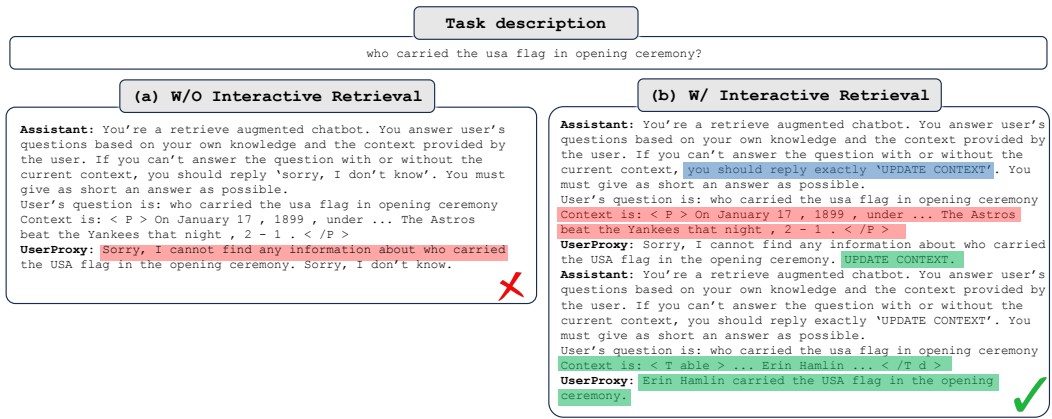

Figure 8: Retrieval-augmented Chat without (W/O) and with (W/) *interactive retrieval*.

An example (Figure 8) from the NQ dataset showcases the advantages of *interactive retrieval* feature: *"who carried the usa flag in opening ceremony"*. When answering this question, the context with the highest similarity to the question embedding does not contain the required information to answer the question. As a result, the LLM assistant (GPT-3.5-turbo) replies *"Sorry, I cannot find any information about who carried the USA flag in the opening ceremony. UPDATE CONTEXT."* With the unique and innovative ability to update context in Retrieval-augmented Chat, the user proxy agent automatically updates the context and sends it to the assistant agent again. After this process, the agent is able to generate the correct answer to the question.

In addition, we conduct an experiment using the same prompt as illustrated in (Adlakha et al., 2023) to investigate the advantages of `AutoGen` *W/O interactive retrieval*. The F1 score and Recall for the first 500 questions are 23.40% and 62.60%, respectively, which are nearly identical to the results reported in Figure 4b. Consequently, we assert that `AutoGen` *W/O interactive retrieval* outperforms *DPR* due to differences in the retrievers employed. Specifically, we use a straightforward vector search retriever with the *all-MiniLM-L6-v2* model for embeddings. Furthermore, we conduct an analysis of the number of LLM calls in experiments involving both `AutoGen` and `AutoGen` *W/O interactive retrieval*, revealing that approximately 19.4% of questions in the Natural Questions dataset trigger an "Update Context" operation, resulting in additional LLM calls.

**Scenario 2: Code generation leveraging latest APIs from the codebase.** In this case, the question is *"How can I use FLAML to perform a classification task and use spark to do parallel training? Train 30 seconds and force cancel jobs if time limit is reached."*. FLAML (v1) (Wang et al., 2021) is an open-source Python library for efficient AutoML and tuning. It was open-sourced in December 2020, and is included in the training data of GPT-4. However, the question requires the use of Spark-related APIs, which were added in December 2022 and are not included in the GPT-4 training data. As a result, the original GPT-4 model is unable to generate the correct code, as it lacks knowledge of Spark-related APIs. Instead, it creates a non-existent parameter, $spark$, and sets it to 'True'. However, with Retrieval-augmented Chat, we provide the latest reference documents as context. Then, GPT-4 generates the correct code blocks by setting $use\_spark$ and $force\_cancel$ to 'True'.

A3: DECISION MAKING IN TEXT WORLD ENVIRONMENTS

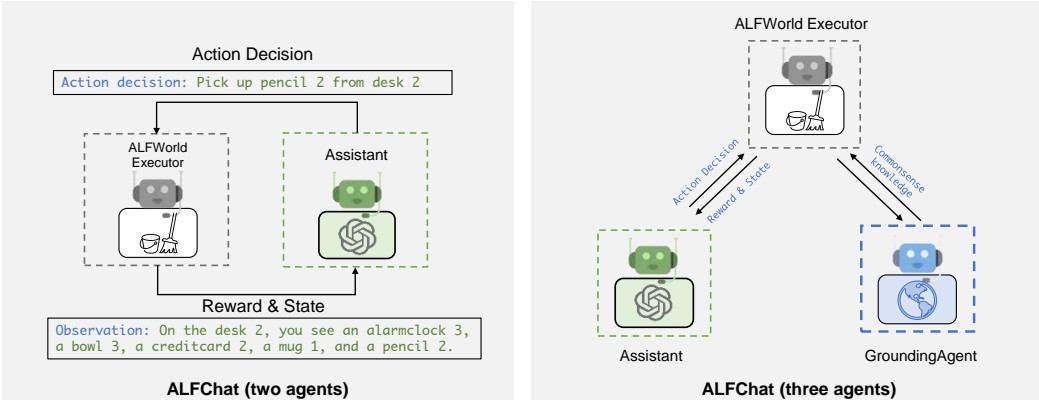

Figure 9: We use `AutoGen` to solve tasks in the ALFWorld benchmark which contains household tasks described in natural language. We propose two designs: A two-agent design where the assistant agent suggests the next step and the Executor executes actions and provides feedback. The three-agent design adds a grounding agent that supplies commonsense facts to the assistant.

ALFWorld (Shridhar et al., 2021) is a synthetic language-based interactive decision-making task. It comprises textual environments that aim to simulate real-world household scenes. Given the high-level goal (e.g., putting a hot apple in the fridge) and the description of the household environment, the agent needs to explore and interact with the simulated household environment through a textual interface. A typical task environment contains various types of locations and could take more than 40 steps to finish, which highlights the need for agents to decompose the goal into subtasks and track them down one by one, meanwhile exploring the environments effectively.

**Detailed Workflow.** We first propose a straightforward design to tackle the benchmark. As illustrated in Figure 9, the system contains two agents. The first assistant agent generates plans to solve the tasks and make action decisions accordingly. The second executor agent is tailored specifically for ALFWorld, which performs action according to assistance and provides feedback from the household environment. Due to the strict format requirements for the output format, we use BLEU metric to evaluate the similarity of the output to all valid action options. The option with the highest similarity will be chosen as the action for this round.

One major challenge encompassed in ALFWorld is commonsense reasoning. The agent needs to extract patterns from the few-shot examples provided and combine them with the agent's general knowledge of household environments to fully understand task rules. More often than not, the assistant tends to neglect some basic knowledge of the household environment. Thanks to the easy-to-implement multi-agent conversational feature of `AutoGen`, it is easy to enhance the assistant agent's reasoning ability by adding a new grounding agent to provide commonsense facts for the decision-making agent's reference. By scrutinizing the failed attempts and summarizing failure reasons, we obtained a holistic understanding of the common sense knowledge that the assistant agent lacks. Then we set a grounding agent to provide this general knowledge when the task begins and whenever the assistant outputs the same action three times in a row. This ensures the assistant takes this commonsense knowledge into consideration and prevents it from getting stuck in outputting the same content or constantly apologizing.

We compare the system's performance with ReAct, which viewed ALFWorld as a text-completion task. ReAct (Yao et al., 2022) is a few-shot prompting technique that interleaves reasoning and acting, allowing for greater synergy between the two and significantly improved performance on both language and decision-making tasks. We integrate ReAct into `AutoGen` by modifying the prompts in a conversational manner. Following ReAct, we employ a two-shot setting. The few-shot prompts are from the corresponding repository. As shown in Table 3. The two-agent design levels the performance of ReAct, while three-agent design significantly outperforms ReAct. We presume

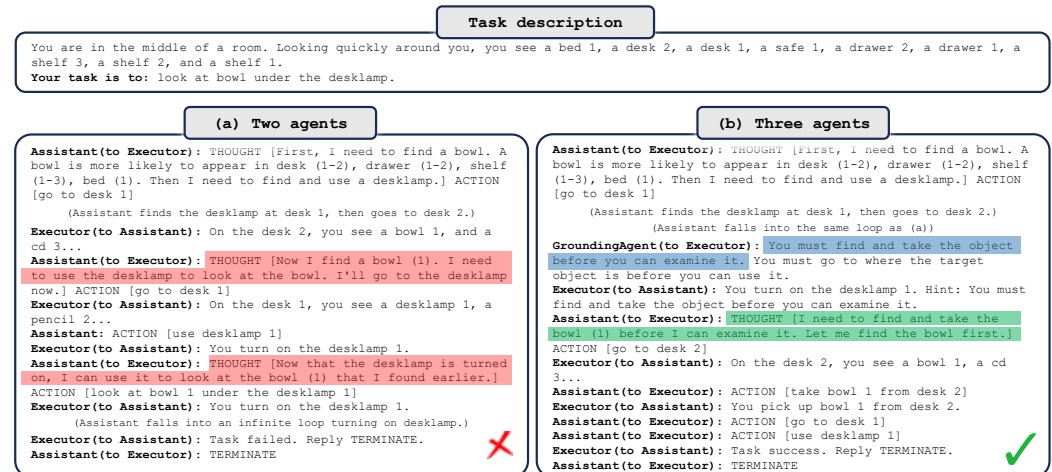

Figure 10: Comparison of two designs, (a) Two-agent design that incorporates interaction between `AssistantAgent` and `UserProxyAgent`, (b) Three-agent design that adds a `GroundingAgent` that serves as a knowledge source. For simplicity, we omit the in-context examples and part of the exploration trajectory and only show parts contributing to the failure/success of the attempt.

| Method | Pick | Clean | Heat | Cool | Look | Pick 2 | All |
|---|---|---|---|---|---|---|---|
| ReAct (avg) | 63 | 52 | 48 | 71 | 61 | 24 | 54 |
| ALFChat (2 agents)(avg) | 61 | 58 | 57 | 67 | 50 | 19 | 54 |
| ALFChat (3 agents)(avg) | 79 | 64 | 70 | 76 | 78 | 41 | 69 |
| ReAct (best of 3) | 75 | 62 | 61 | 81 | 78 | 35 | 66 |
| ALFChat (2 agents)(best of 3) | 71 | 61 | 65 | 76 | 67 | 35 | 63 |
| AFLChat (3 agents)(best of 3) | 92 | 74 | 78 | 86 | 83 | 41 | 77 |

Table 3: Comparisons between ReAct and ALFChat on ALFWorld benchmark. For each task, we report the success rate out of 3 attempts. Success rate stands for the number of tasks successfully completed by the agent divided by the total number of tasks. The results show that by adding a grounding agent, the assistant agent leverages necessary commonsense knowledge into reasoning and improves task success rate significantly.

that the performance discrepancy is caused by the inherent difference between dialogue-completion and text-completion tasks. On the other hand, introducing a grounding agent as a knowledge source remarkably advances performance on all types of tasks.

**Case study**. Figure 10 exemplifies how a three-agent design eliminates one root cause for failure cases. Most of the tasks involve taking an object and then doing something to it (e.g. finding a vase and putting it in a safe place). When there is no grounding agent, the assistant frequently links finding an object and taking it away as equivalent Figure 10a). This leads to most of the failure cases in 'pick' and 'look' type tasks. With the introduction of a grounding agent, the assistant can jump out of the loop and successfully complete the task.

**Takeaways.** By decoupling ALFWorld agent-text interaction into assistant and executor, our design offered both simplicity and modularity. We introduced a grounding agent to serve as an external commonsense knowledge source, which significantly enhanced the assistant's ability to make informed decisions. This proves providing necessary commonsense facts to the decision-making agent can assist it in making more informed decisions, thus boosting task success rate effectively.

A4: MULTI-AGENT CODING

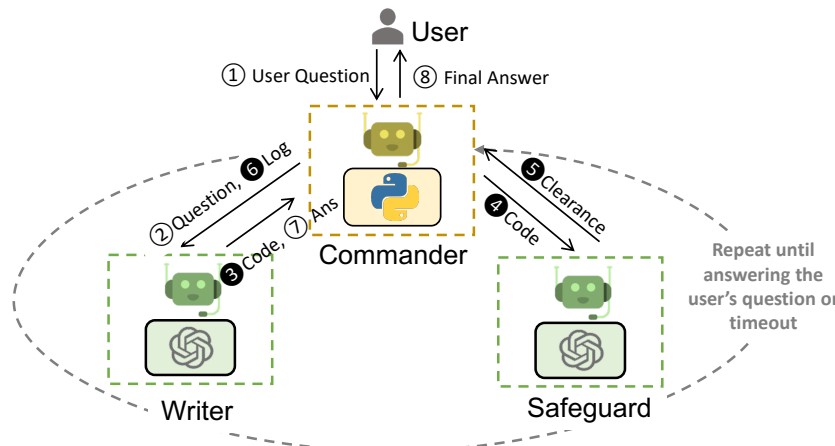

Figure 11: Our re-implementation of *OptiGuide* with `AutoGen` streamlining agents' interactions. The Commander receives user questions (e.g., What if we prohibit shipping from supplier 1 to roastery 2?) and coordinates with the Writer and Safeguard. The Writer crafts the code and interpretation, the Safeguard ensures safety (e.g., not leaking information, no malicious code), and the Commander executes the code. If issues arise, the process can repeat until resolved. Shaded circles represent steps that may be repeated multiple times.

**Detailed Workflow.** The workflow can be described as follows. The user agent initiates the interaction by posing the user's question, such as "What if we prohibit shipping from supplier 1 to roastery 2?", marked by ① to the Commander agent. The Commander manages and coordinates with two LLM-based assistant agents: the Writer and the Safeguard. Apart from directing the flow of communication, the Commander has the responsibility of handling memory tied to user interactions. This capability enables the Commander to capture and retain valuable context regarding the user's questions and their corresponding responses. Such memory is subsequently shared across the system, empowering the other agents with context from prior user interactions and ensuring more informed and relevant responses.

In this orchestrated process, the Writer, who combines the functions of a "Coder" and an "Interpreter" as defined in (Li et al., 2023a), will craft code and also interpret execution output logs. For instance, during code writing (② and ❸), the Writer may craft code "model.addConstr(x['supplier1', 'roastery2'] == 0, 'prohibit')" to add an additional constraint to answer the user's question.

After receiving code, the Commander will communicate with the Safeguard to screen the code and ascertain its safety (❹); once the code obtains the Safeguard's clearance, marked by ❺, the Commander will use external tools (e.g., Python) to execute the code, and request the Writer to interpret the execution results for the user's question (❻ and ⑦). For instance, the writer may say "if we prohibit shipping from supplier 1 to roastery 2, the total cost would increase by 10.5%." Bringing this intricate process full circle, the Commander furnishes the user with the concluding answer (⑧).

However, if at a point there is an exception - either a security red flag raised by Safeguard (in ❺) or code execution failures within Commander, the Commander redirects the issue back to the Writer with essential information in logs (❻). So, the process from ❸ to ❻ might be repeated multiple times, until each user query receives a thorough and satisfactory resolution or until the timeout. This entire complex workflow of multi-agent interaction is elegantly automated via `AutoGen` agent chat.

The core workflow code for OptiGuide was reduced from over 430 lines to 100 lines using `AutoGen`, leading to significant productivity improvement. The new agents are customizable, conversable, and can autonomously manage their chat memories. This consolidation allows the coder and interpreter roles to merge into a single "Writer" agent, resulting in a clean, concise, and intuitive implementation that is easier to maintain.

**Manual Evaluation Comparing ChatGPT + Code Interpreter and OptiGuide.** ChatGPT + Code Interpreter is unable to run code with private or customized dependencies (e.g., Gurobi), which means users need to have engineering expertise and manually handle multiple steps, disrupting the workflow and increasing the chance for mistakes. If users lack access or expertise, the burden falls on supporting engineers, increasing their on-call time.

We carried out a user study that juxtaposed OpenAI's ChatGPT coupled with a Code Interpreter against `AutoGen`-based OptiGuide. The study focused on a coffee supply chain scenario, and an expert Python programmer with proficiency in Gurobi participated in the test. We evaluated both systems based on 10 randomly selected questions, measuring time and accuracy. While both systems answered 8 questions correctly, the Code Interpreter was significantly slower than OptiGuide because the former requires more manual intervention. On average, users need to spend 4 minutes and 35 seconds to solve problems with the Code Interpreter, with a standard deviation of approximately 2.5 minutes. In contrast, OptiGuide's average problem-solving time was around 1.5 minutes, most of which was spent waiting for responses from the GPT-4 model. This indicates a 3x saving on the user's time with OptiGuide.

While using ChatGPT + Code Intepreter, users had to read through code and instructions to know where to paste the code snippets. Additionally, running the code involves downloading it and executing it in a terminal, a process that was both time-consuming and prone to errors. The response time from Code Interpreter is also slower, because it generates lots of tokens to read the code, read the variables line-by-line, perform chains of thought analysis, and then produce the final answer code. In contrast, `AutoGen` integrates multiple agents to reduce user interactions by 3 - 5 times on average as reported in Table 4, where we evaluated our system with 2000 questions across five OptiGuide applications and measured how many prompts the user need to type.

Table 4: Manual effort saving in OptiGuide (W/ GPT-4), while preserving the same coding performance. Data displayed include both the mean and standard deviations (indicated in parentheses).

| Dataset | netflow | facility | tsp | coffee | diet |
|---|---|---|---|---|---|
| Saving Ratio | 3.14x (0.65) | 3.14x (0.64) | 4.88x (1.71) | 3.38x (0.86) | 3.03x (0.31) |

Table 13 and 15 provide a detailed comparison of user experience with ChatGPT+Code Interpreter and `AutoGen`-based OptiGuide. ChatGPT+Code Interpreter is unable to run code with private packages or customized dependencies (such as Gurobi); as a consequence ChatGPT+Code Interpreter requires users to have engineering expertise and manually handle multiple steps, disrupting the workflow and increasing the chance for mistakes. If customers lack access or expertise, the burden falls on supporting engineers, increasing their on-call time. In contrast, the automated chat by `AutoGen` is more streamlined and autonomous, integrating multiple agents to solve problems and address concerns. This results in a 5x reduction in interaction and fundamentally changes the overall usability of the system. A stable workflow can be potentially reused for other applications or to compose a bigger one.

**Takeaways:** The implementation of the multi-agent design with `AutoGen` in the OptiGuide application offers several advantages. It simplified the Python implementation. It fostered a mixture of collaborative and adversarial problem-solving environment, with the Commander and Writer working together while the Safeguard acts as a virtual adversarial checker. This setup allows for proper memory management, as the Commander maintains memory related to user interactions, providing context-aware decision-making. Additionally, role-playing ensures that each agent's memory remains isolated, preventing shortcuts and hallucinations.

A5: DYNAMIC GROUP CHAT

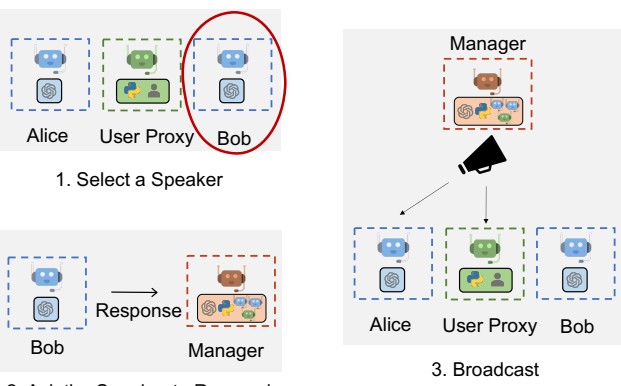

Figure 12: A5: Dynamic Group Chat: Overview of how `AutoGen` enables dynamic group chats to solve tasks. We use a special agent called the Manager that repeats the following three steps– select a single speaker (in this case Bob), ask the speaker to respond, and broadcast the selected speaker's message to all the other agents.

To validate the necessity of multi-agent dynamic group chat and the effectiveness of the role-play speaker selection policy, we conduct a pilot study comparing a four-agent dynamic group chat system with two possible alternatives across 12 manually crafted complex tasks. An example task is *"How much money would I earn if I bought 200 $AAPL stocks at the lowest price in the last 30 days and sold them at the highest price? Save the results into a file."* The four-agent group chat system was composed of the following group members: a user proxy to take human inputs, an engineer to write code and fix bugs, a critic to review code and provide feedback, and a code executor for executing code. One of the possible alternatives is a two-agent system involving an LLM-based assistant and a user proxy agent, and another alternative is another group chat system with the same group members but a task-based speaker selection policy. In the task-based speaker selection policy, we just append role information, chat history, and the next speaker's task into a single prompt. Through the pilot study, we observe that compared with a task-style prompt, utilizing a role-play prompt in dynamic speaker selection often leads to more effective consideration of both conversation context and role alignment during the process of generating the subsequent speaker and consequently a higher success rate and fewer LLM calls.

Table 5: Number of successes on the 12 tasks (higher the better).

| Model | Two Agent | Group Chat | Group Chat with a task-based speaker selection policy |
|---|---|---|---|
| GPT-3.5-turbo | 8 | **9** | 7 |
| GPT-4 | 9 | **11** | 8 |

Table 6: Average # LLM calls and number of termination failure for each task (lower the better).

| Model | Two Agent | Group Chat | Group Chat with a task-based speaker selection policy |
|---|---|---|---|
| GPT-3.5-turbo | 9.9, 9 | 5.3, 0 | 4, 0 |
| GPT-4 | 6.8, 3 | 4.5, 0 | 4, 0 |

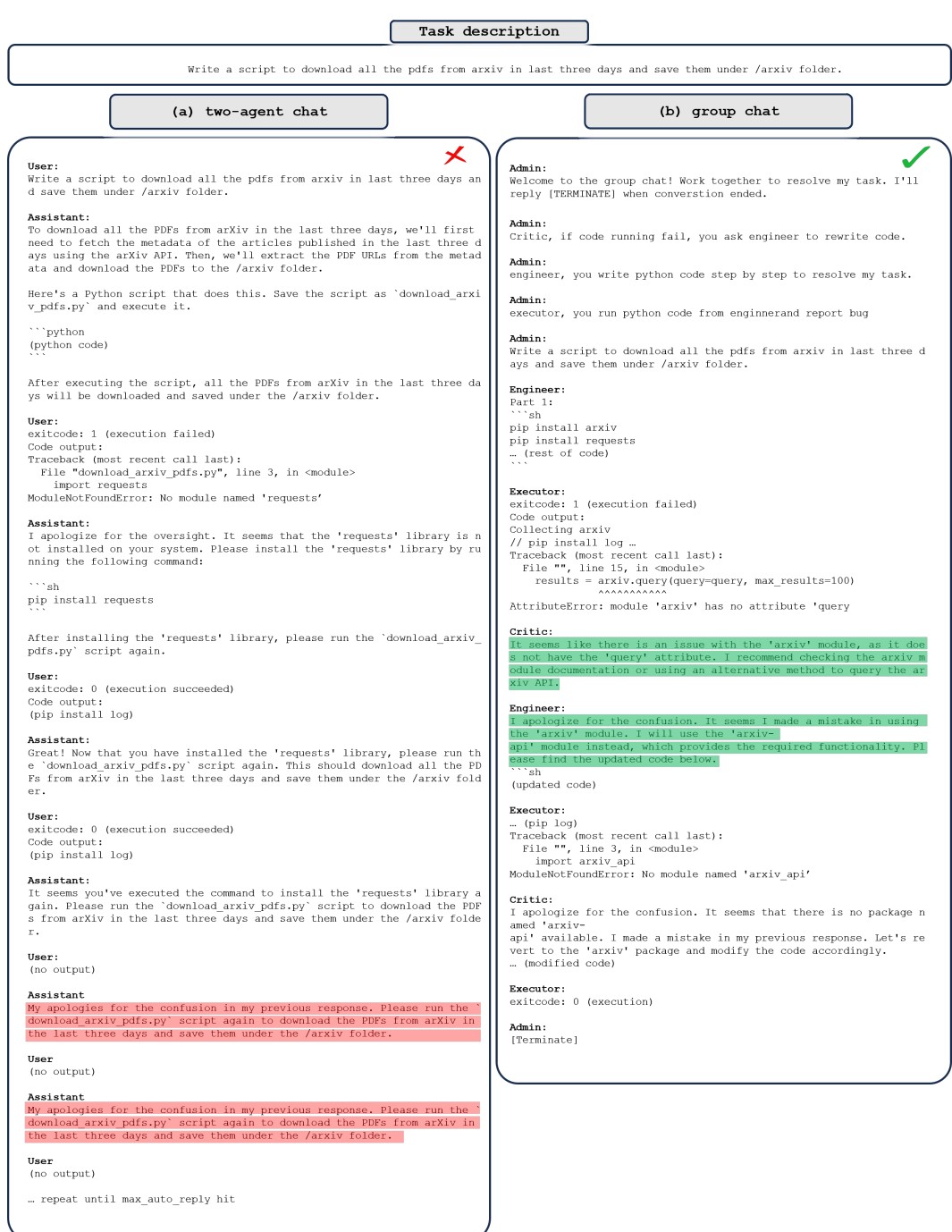

Figure 13: Comparison of two-agent chat (a) and group chat (b) on a given task. Group chat resolves the task successfully with a smoother conversation, while the two-agent chat fails on the same task and ends with repeated conversation.

A6: CONVERSATIONAL CHESS

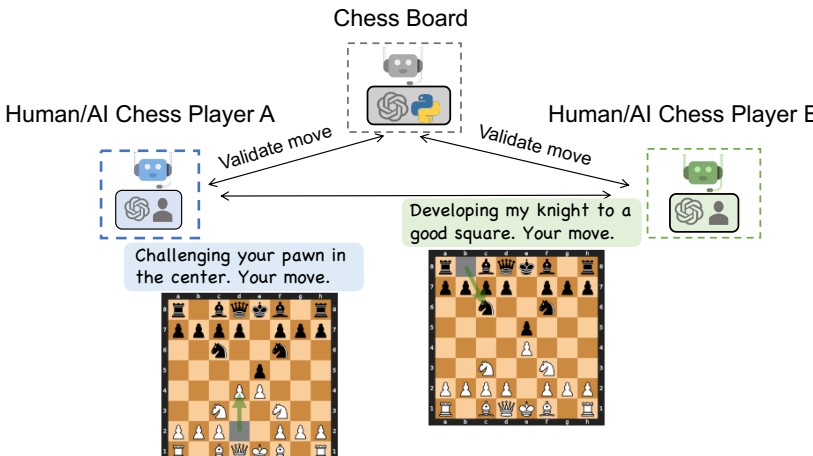

Figure 14: A6: Conversational Chess: Our conversational chess application can support various scenarios, as each player can be an LLM-empowered AI, a human, or a hybrid of them. Here, the board agent maintains the rules of the game and supports the players with information about the board. Players and the board agent all use natural language for communication.

In Conversational Chess, each player is an `AutoGen` agent and can be either powered by a human or an AI. A third party that is a board agent is built to support players with information about the board and ground players' moves to a legal chess move. Figure 14 shows the scenarios supported by Conversational Chess: AI/human vs. AI/human, and how players and a board agent interact. It opens up opportunities for social interaction and allows players to express their moves creatively, such as using jokes, meme references, and character-playing, therefore making chess games more entertaining to players as well as observers (Figure 15 shows an example of conversational chess).

To realize these scenarios, we constructed a player agent with LLM and human as back-end options. When human input is enabled, before sending the input to the board agent, it first asks the human player to input the message that contains the move and anything else the player wants to say (such as a witty comment). If human input is skipped or disabled, LLM is used to generate the message. The board agent is implemented with a custom reply function, which uses an LLM to parse the natural language input into a legal move in a structured format (e.g., UCI), and then pushes the move to the board. If the move is not legitimate, the board agent will reply with an error. Then the player agent needs to resend a message to the board agent until the move is legal. Once the move is successfully pushed, the player agent sends the message to the opponent. As in Figure 15, the conversation between AI players can be natural and entertaining. When the player agents use LLM to generate a message, it makes use of the board state and the error message from the board agent. That helps reduce the chance of hallucinating an invalid move. The chat between one player agent and the board agent is invisible to the other player agent, which keeps the messages used in chat completion well managed.

There are two notable benefits of using `AutoGen` to implement Conversational Chess. Firstly, the agent design in `AutoGen` enables the natural creation of objects and their interactions needed in our chess game. This makes development easy and intuitive. For example, the isolation of chat messages makes it easy to make a proper LLM chat completion inference call. Secondly, `AutoGen` greatly simplifies the implementation of agent behaviors using composition. Specifically, we used the `register_reply` method supported by `AutoGen` agents to instantiate player agents and a board agent with custom reply functions. Restricting the extension work needed in a single point (the reply function) simplifies the reasoning, development, and maintenance.

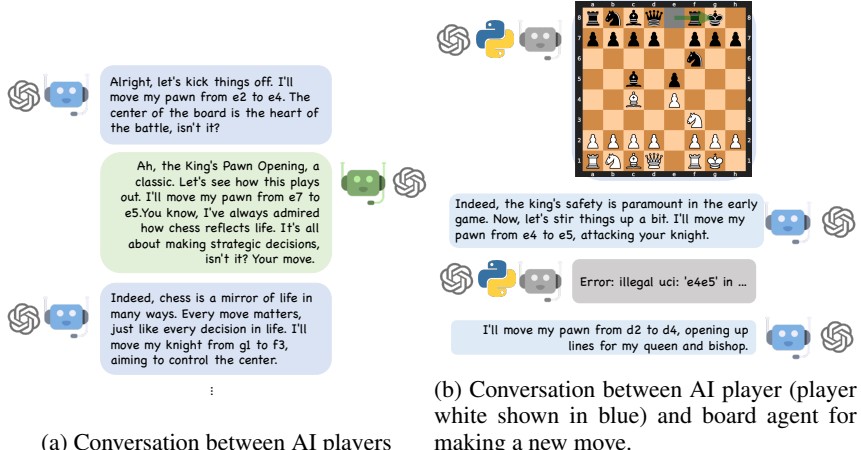

(a) Conversation between AI players

(b) Conversation between AI player (player white shown in blue) and board agent for making a new move.

Figure 15: Example conversations during a game between two AI players.

To show the effect brought by this board agent, we provide a demonstration of conversational chess without a board agent Figure 16. In this demonstration, instead of using an additional board agent to do grounding, the system uses prompting to do grounding by including this sentence "You should make sure both you and the opponent are making legal moves." to the system message of both players.

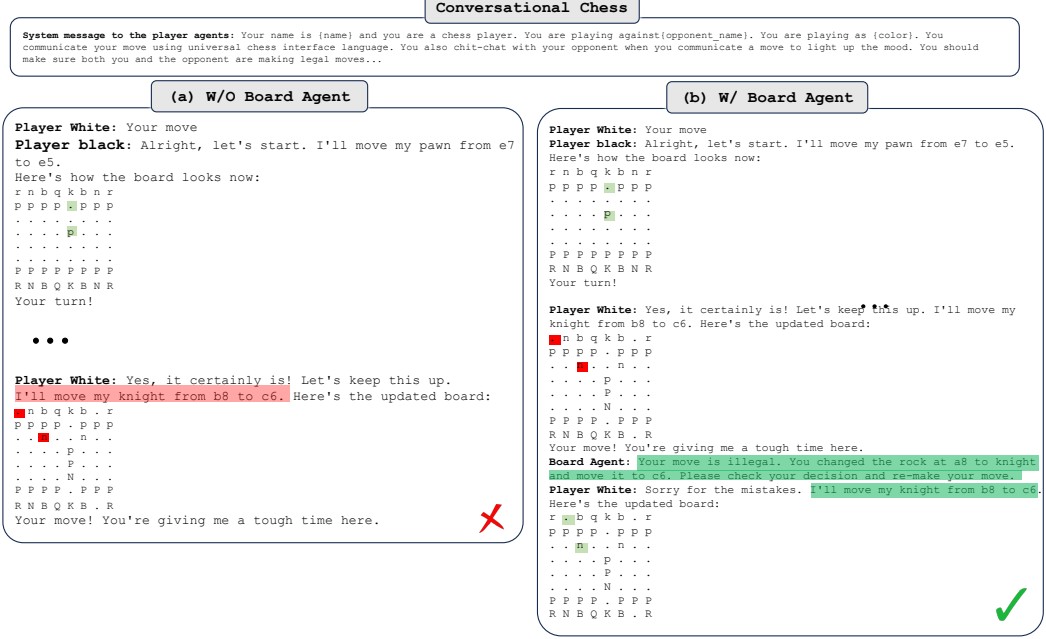

Figure 16: Comparison of two designs–(a) without a broad agent, and (b) without a board agent–in Conversational Chess

A7: ONLINE DECISION MAKING FOR BROWSER INTERACTIONS

Figure 17: We use `AutoGen` to build MiniWobChat to solve tasks in the MiniWob++ benchmark. MiniWobChat consists of two agents: An assistant agent and an executor agent. The assistant agent suggests actions to manipulate the browser, and the executor executes the suggested action and returns reward/feedback. The assistant agent records feedback and continues until the feedback indicates task success/failure.

In practice, many applications demand the presence of agents capable of interacting with environments and making decisions in an online context, such as game playing (Mnih et al., 2013; Vinyals et al., 2017), web interactions (Liu et al., 2018; Shi et al., 2017) and robot manipulations (Shen et al., 2021). With the multi-agent conversational framework in `AutoGen`, it is easy to decompose the automatic agent-environment interactions and the development of a decision-making agent by constructing an *executor* agent responsible for handling the interaction with the environment, therefore splitting out the decision-making part to other agents. Such a decomposition allows developers to reuse the decision-making agent for new tasks with minimal effort rather than building a specialized decision-making agent for every new environment.

**Workflow.** We demonstrate how to use `AutoGen` to build a working system to handle such scenarios with the MiniWoB++ benchmark (Shi et al., 2017). MiniWoB++ comprises browser interaction tasks that involve utilizing mouse and keyboard actions to interact with browsers. The ultimate objective of each task is to complete tasks described concisely in natural language, such as "expand the web section below and click the submit button." Solving these tasks typically requires a sequence of web manipulation actions rather than one single action, and making action decisions at each time step requires access to the web status (in the form of HTML code) online. For the example above, clicking the submit button requires getting the status of the web status after expanding the web section. We designed a straightforward two-agent system named MiniWobChat using `AutoGen` as shown in Figure 17. The assistant agent is an instance of the built-in `AssistantAgent` and is responsible for making action decisions for the given task. The second agent, the executor agent, is a customized `UserProxyAgent`, which is responsible for interacting with the benchmark by executing the actions suggested by the `AssistantAgent` and returning feedback.

To assess the developed working system's performance, we make a comparison with RCI (Kim et al., 2023), a recent solution for MiniWoB++ benchmark that employs a set of self-critiquing prompts and has achieved the state-of-the-art performance. In our evaluation, we use all available tasks in RCI official code with varying degrees of difficulty to conduct a comprehensive analysis against MiniWobChat. Figure 18 illustrates that MiniWobChat achieves competitive performance in this evaluation[8]. Specifically, among the 49 available tasks, MiniWobChat achieves a success rate of 52.8%, which is only 3.6% lower than RCI, a method specifically designed for the MiniWob++ benchmark. It is worth noting that in most tasks, the difference between the two methods is mirrored as shown in Figure 18. If we consider 0.1 as a success rate tolerance for each task, i.e., two methods that differ within 0.1 are considered to have the same performance, both two methods achieve better

---

[8]We report the results of RCI by running its official code with default settings.

performance than the other on the same number of tasks. For illustration purpose, we provide a case analysis in Table 7 on four typical tasks.

Additionally, we also explored the feasibility of using Auto-GPT for handling the same tasks. Auto-GPT faces challenges in handling tasks that involve complex rules due to its limited extensibility. It provides an interface for setting task goals using natural language. However, when dealing with the MiniWob++ benchmark, accurately instructing Auto-GPT to follow the instructions for using Miniwob++ is challenging. There is no clear path to extend it like the two-agent chat by `AutoGen`.

**Takeaways:** For this application, `AutoGen` stood out as a more user-friendly option, offering modularity and programmability: It streamlined the process with autonomous conversations between the assistant and the executor and provided readily available solutions for agent-environment interactions. The built-in `AssistantAgent` was directly reusable and presented strong performance without customization. Moreover, the decoupling of the execution and assistant agent ensures that modifications to one component do not adversely impact the other. This convenience makes maintenance and future updates straightforward.

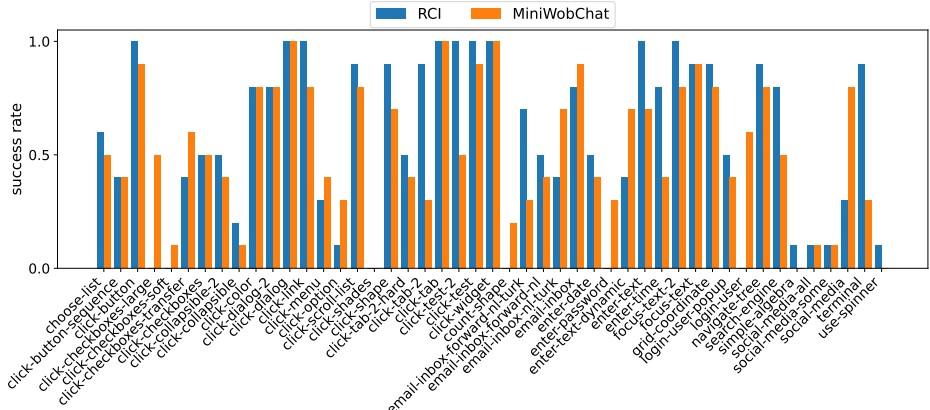

Figure 18: Comparisons between RCI (state-of-the-art prior work) and MiniWobChat on the Mini-Wob++ benchmark. we use all available tasks in RCI official code with varying degrees of difficulty to conduct comprehensive comparisons. For each task, we reported the success rate for ten different instances. The results show that MiniWobChat achieves comparable performance with RCI. If we consider 0.1 as a success rate tolerance for each task, both methods achieve better performance than the other on the same number of tasks.

Table 7: Cases analysis on four typical tasks from MiniWob++ benchmark.

| | Correctness | Main failure reason |
|---|---|---|
| click-dialog | `AutoGen`: 10/10 | N/A. |
| | RCI: 10/10 | N/A. |
| click-checkboxes-large | `AutoGen`: 5/10 | `AssistantAgent` provides actions with infeasible characters. |
| | RCI: 0/10 | RCI performs actions that are out of its plan. |
| count-shape | `AutoGen`: 2/10 | `AssistantAgent` provide actions with redundant content that can not convert to actions in the benchmark. |
| | RCI: 0/10 | RCI provides a wrong plan in most cases. |
| use-spinner | `AutoGen`: 0/10 | `AssistantAgent` return actions out of its plan. |
| | RCI: 1/10 | RCI provides a wrong plan in most cases. |

# E    EXAMPLE OUTPUTS FROM APPLICATIONS

In this section, we include example outputs from the following applications and systems:

- Application A1: autonomous solving process of one problem on: ChatGPT + Plugin (Table 8), `AutoGen` (Table 9), LangChain ReAct (Table 10), Auto-GPT (Table 11), Multi-Agent Debate (Table 12), ChatGPT + Code Interpreter (Table 14).
- Application A4: an OptiGuide problem with `AutoGen` (Table 15) and ChatGPT + Code Interpreter (Table 13).
- Application A1: Preliminary evaluation on several alternative multi-agent frameworks which shows that they should be excluded from quantitative comparison: BabyAGI (Table 16), CAMEL (Table 17), MetaGPT (Table 18).

Table 8: Application A1. Scenario 1: autonomous math solving. ChatGPT+ Plugin (Wolfram Alpha) on the first qualitative example. The response from Wolfram contains the correct answer but ChatGPT chose the wrong answer.

|  | Action | ChatGPT + Plugin |
|---|---|---|
| 👤 | Input | Completely simplify and rationalize the denominator: $$\frac{\sqrt{160}}{\sqrt{252}} \times \frac{\sqrt{245}}{\sqrt{108}}$$ |
| 🤖 | | REQUEST TO WOLFRAM: 

```
1  {
2    "input": "simplify (sqrt(160)/sqrt(252))*
3    (sqrt(245)/sqrt(108))"
4  }
```

RESPONSE FROM WOLFRAM

```
1  Query:
2  "simplify (sqrt(160)/sqrt(252))*(sqrt(245)/sqrt(108))"
3
4  Input interpretation:
5  simplify | sqrt(160)/sqrt(252)x sqrt(245)/sqrt(108)
6
7  Results:
8  (5 sqrt(42))/27
9
10 (5 sqrt(14/3))/9
11
12 Decimal approximation:
13 1.2001371663718259686974013770533327143898526494552...
14
15 Wolfram|Alpha website result for "simplify
16 (sqrt(160)/sqrt(252))*(sqrt(245)/sqrt(108))":
17 [link]
``` |
| 🤖 | | The completely simplified and rationalized form of the given expression is: $$\frac{5\sqrt{14/3}}{9}$$ In decimal form, this is approximately 1.20013716637182596869740137705333271 4... |

Table 9: Application A1. Scenario 1: autonomous math solving. `AutoGen` on the first qualitative example. `AutoGen` solves the problem correctly.

| | Action | **AutoGen** |
|---|---|---|
| 👤 | Init chat | Completely simplify and rationalize the denominator: $$\frac{\sqrt{160}}{\sqrt{252}} \times \frac{\sqrt{245}}{\sqrt{108}}$$ |
| 🤖 | | We can simplify and rationalize this denominator using Python's symbolic mathematics library 'sympy'. Here is the Python code to resolve this: 

 ```python
from sympy import sqrt, Rational

# Define the entire fraction as a rational number
fraction = (sqrt(160)/sqrt(252))*(sqrt(245)/sqrt(108))

# Simplify the fraction
simplified = fraction.simplify()

# Print the simplified fraction
print(simplified)
``` 

 This code performs the requested simplification by constructing the fraction using the 'Rational()' function from the 'sympy' module, which automatically simplifies fractions when possible. The 'sqrt()' function from the 'sympy' module enables square root calculations. Finally, the 'simplify()' function simplifies expressions when possible. |
| 👤 | Auto Reply | exitcode: 0 (execution succeeded) Code output: 5*sqrt(42)/27 |
| 🤖 | | TERMINATE |

Table 10: Application A1. Scenario 1: autonomous math solving. LangChain ReAct on the first qualitative example. The code generated does not match the reasoned plan. The code returns a decimal number.

| | Action | **LangChain ReAct** |
|---|---|---|
| 👤 | Input | Completely simplify and rationalize the denominator: $$\frac{\sqrt{160}}{\sqrt{252}} \times \frac{\sqrt{245}}{\sqrt{108}}$$ |
| 🤖 | | Entering new AgentExecutor chain... 
 To simplify this expression, I will first simplify each square root individually, then multiply the fractions as normal. The result should be a rational number. Action: Python_REPL Action Input: 

 ```python
import math

# Calculate square roots
sqrt_160 = math.sqrt(160)
sqrt_252 = math.sqrt(252)
sqrt_245 = math.sqrt(245)
sqrt_108 = math.sqrt(108)

# Multiply fractions
numerator = sqrt_160 * sqrt_245
denominator = sqrt_252 * sqrt_108
result = numerator / denominator

print(result)
``` 

 Observation: 1.200137166371826 
 Thought: I now know the final answer Final Answer: The simplified and rationalized form of the expression is approximately 1.200137166371826. 
 Finished chain. |

Table 11: Application A1. Scenario 1: autonomous math solving. Auto-GPT on the first qualitative example. We abort since the code never prints the result and the system does not fix it after 2 turns. Auto-GPT is much more verbose than other frameworks. We shorten some unnecessary content.

| | Action | Auto-GPT |
|---|---|---|
| | System | Welcome back! Would you like me to return to being MathSolverGPT? Asking user via keyboard... Continue with the last settings?
Name: MathSolverGPT
Role: an intelligent math-solving AI that assists students, professionals,...
Goals: ['Provide step-by-step solutions to math problems, ensuring clarity and understanding for the user...
Continue (y/n): |
| 👤 | Human | y |
| | System | MathSolverGPT has been created with the following details: Name: MathSolverGPT
Role: an intelligent math-solving AI that assists students, professionals, and enthusiasts in solving a wide range of math problems, from basic arithmetic to advanced calculus and beyond.
Goals: - Provide step-by-step solutions to math problems, ensuring clarity and understanding for the user. - Offer explanations and insights into the underlying concepts and principles behind the solutions, fostering a deeper understanding of mathematics. - Handle a variety of math problem types, including algebra, geometry, trigonometry, calculus, and more. ... |
| ♟ | | MATHSOLVERGPT THOUGHTS: Without a specific task or question, it's difficult to determine exactly one command to use. The user hasn't provided any context or problem to solve.
REASONING: The user's request is too vague. They haven't specified what they want me to do or what problem they want me to solve.
PLAN: - Ask the user for more information or a specific task.
CRITICISM: I should have asked the user for more information or a specific task in the first place.
SPEAK: Could you please provide more information or specify a task?
NEXT ACTION: COMMAND = ARGUMENTS = { } |
| | System | Enter 'y' to authorise command, 'y -N' to run N continuous commands, 'n' to exit program, or enter feedback for MathSolverGPT... Asking user via keyboard... |
| 👤 | Human | Completely simplify and rationalize the denominator:

$$\frac{\sqrt{160}}{\sqrt{252}} \times \frac{\sqrt{245}}{\sqrt{108}}$$ |
| ♟ | | MATHSOLVERGPT THOUGHTS: To solve this problem, I need to simplify and rationalize the denominator of the given expression. This involves simplifying the square roots and then multiplying the fractions. I can use Python's math library to calculate the square roots and perform the multiplication.
REASONING: The problem involves simplifying square roots and multiplying fractions. Python's math library provides functions for these operations.
PLAN: - Write a Python script to calculate the square roots and multiply the fractions. - Execute the script to get the simplified expression.
CRITICISM: I need to ensure that I correctly calculate the square roots and perform the multiplication. I should also make sure to rationalize the denominator, as the problem specifies.
SPEAK: Let's simplify and rationalize the denominator of the given expression.
NEXT ACTION: COMMAND = execute_python_code ARGUMENTS =

```\n1 {'code': 'import math\n\n# Calculate the square roots\nsqrt_160 = math.sqrt(160)\n    sqrt_252 = math.sqrt(252)\nsqrt_245 = math.sqrt(245)\nsqrt_108 = math.sqrt\n    (108)\n\n# Multiply the fractions\nresult = (sqrt_160 / sqrt_252) * (sqrt_245\n    / sqrt_108)\n\n# Simplify the result\nsimplified_result = math.simplify(result\n    )\n\nsimplified_result', 'name': 'simplify_expression'}\n``` |
| 👤 | Human | y |
| | System | -=-=-=-=-=-= COMMAND AUTHORISED BY USER -=-=-=-=-=-=-=
Executing python file '/app/autogpt/workspa...
SYSTEM: Command execute_python_code returned: Error: Traceback (most recent call last): File "...", line 13, in ¡module¿ simplified_result = math.simplify(result) AttributeError: module 'math' has no attribute 'simplify' |
| ♟ | | MATHSOLVERGPT THOUGHTS: The error message indicates that the 'math' module in Python does not have a 'simplify' function. I need to use the 'sympy' library instead, which provides a function for simplifying expressions.
REASONING: The 'math' module in Python does not provide a function for simplifying expressions. The 'sympy' library, on the other hand, does provide such a function.
PLAN: - Rewrite the Python script to use the 'sympy' library instead of the 'math' module. - Execute the script to get the simplified expression.
CRITICISM: I should have known that the 'math' module does not provide a function for simplifying expressions. I should have used the 'sympy' library from the start. SPEAK: Let's try again using the 'sympy' library to simplify the expression.

NEXT ACTION: COMMAND = execute_python_code ARGUMENTS =

```\n1 {'code': 'import sympy as sp\n\n# Calculate the square roots\nsqrt_160 = sp.sqrt\n    (160)\nsqrt_252 = sp.sqrt(252)\nsqrt_245 = sp.sqrt(245)\nsqrt_108 = sp.sqrt\n    (108)\n\n# Multiply the fractions\nresult = (sqrt_160 / sqrt_252) * (sqrt_245\n    / sqrt_108)\n\n# Simplify the result\nsimplified_result = sp.simplify(result)\\\n    n\nsimplified_result', 'name': 'simplify_expression'}\n``` |
| 👤 | Human | y |
| | System | SYSTEM: Command execute_python_code returned: |
| ♟ | | MATHSOLVERGPT THOUGHTS: The command execution didn't return any output, which is unusual. It seems there might be an issue with the execution of the Python code. I need to debug this.
REASONING: The Python code execution should have returned a result. The absence of a result suggests that there might be an issue with the code or its execution.
PLAN: - Review the Python code to identify any potential issues. - If necessary, modify the code and execute it again. CRITICISM: I should have ensured that the Python code was correct and would produce a result before executing it. SPEAK: It seems there was an issue with the execution of the Python code. Let's try again.
NEXT ACTION: COMMAND = execute_python_code ARGUMENTS = ... (Exact same code as last) |
| | System | SYSTEM: Command execute_python_code returned: |
| ♟ | | MATHSOLVERGPT THOUGHTS: The Python code execution still isn't returning any output. It's possible that the issue lies with the calculation of the square roots ... |
| 👤 | Human | control + C (Aborted) |

Table 12: Application A1. Scenario 1: autonomous math solving. Multi-Agent Debate on the first qualitative problem. It gives a wrong final answer.

| | Action | AutoGen |
|---|---|---|
| 👤 | Topic | Completely simplify and rationalize the denominator: $$\frac{\sqrt{160}}{\sqrt{252}} \times \frac{\sqrt{245}}{\sqrt{108}}$$ |
| ♣ | | ===== Debate Round-1 ===== |

—— Affirmative side ——
Ok, let's simplify each square root separately. We want to find the simplest square root that can divide 160, 252, 245, and 108.
1. $\sqrt{160}$: 160 can be decomposed as 16*10, where 16 is a perfect square, so $\sqrt{160} = \sqrt{16*10} = \sqrt{16}*\sqrt{10} = 4\sqrt{10}$
2. $\sqrt{252}$: 252 can be decomposed as 36*7, where 36 is a perfect square, so $\sqrt{252} = \sqrt{36*7} = 6\sqrt{7}$
3. $\sqrt{245} = \sqrt{49*5}$, where 49 is a perfect square, so $\sqrt{245} = 7*\sqrt{5}$
4. $\sqrt{108}$: 108 can be decomposed as 36*3, where 36 is a perfect square, so $\sqrt{108} = \sqrt{36*3} = 6\sqrt{3}$
Then we substitute the simplified square roots back into the expression, we have:

$$\frac{\sqrt{160}}{\sqrt{252}} \times \frac{\sqrt{245}}{\sqrt{108}} = \frac{4\sqrt{10}}{6\sqrt{7}} \times \frac{7\sqrt{5}}{6\sqrt{3}} = \frac{4*7}{6*6}\cdot\sqrt{\frac{10*5}{7*3}} = \frac{7}{9}\cdot\sqrt{\frac{50}{21}}$$

We are almost done, but we still need to rationalize the square root part of expression. Let's multiply the square root part by $\sqrt{21}$ at both the top and bottom to get rid of the square root at the denominator:

$$\frac{7}{9}\cdot\sqrt{\frac{50*21}{21*21}} = \frac{7\sqrt{1050}}{189}.$$

Then this is the final answer! The expression

$$\frac{\sqrt{160}}{\sqrt{252}} \times \frac{\sqrt{245}}{\sqrt{108}}$$

can be simplified to $\frac{7\sqrt{1050}}{189}$.
—— Negative side ——
I do disagree with your final answer, but I think we mostly agree on the steps. Your simplifications of the square roots are correct, but you made a mistake in the final simplification part. Let me explain how I would complete the problem:
As stated,

$$\frac{\sqrt{160}}{\sqrt{252}} \times \frac{\sqrt{245}}{\sqrt{108}} = \frac{4\sqrt{10}}{6\sqrt{7}} \times \frac{7\sqrt{5}}{6\sqrt{3}}.$$

This should then simplify to:

$$= \frac{28\sqrt{50}}{36\sqrt{21}} = \frac{7\sqrt{50}}{9\sqrt{21}}.$$

You made a mistake in your calculation because you considered square root of $50/21$, and you should have considered the simplified square root of 50 over the simplified square root of 21. The rationalization would become:

$$= \frac{7\sqrt{50}}{9\sqrt{21}}\cdot\frac{\sqrt{21}}{\sqrt{21}} = \frac{7\sqrt{1050}}{189}.$$

This seems to coincide with your final answer, but it's in fact a very different number, So, the original expression simplifies to $\frac{7\sqrt{1050}}{189}$.
Let's now see if we disagree on this final answer.
—— Moderator ——
"Whether there is a preference": "Yes", "Supported Side": "Negative", "Reason": "Both sides of the debate came to the same final answer. However, the negative side correctly identified a mistake in the affirmative side's calculation, pointing out that the simplification should have been the square root of 50 over the square root of 21, not 50 over 21. The negative side shows a better understanding of the simplification process.", "debate_answer": "$\frac{7\sqrt{1050}}{189}$"
===== Debate Done! =====
—— Base Answer ——
Ok, let's simplify each square root separately. We want to find the simplest square root that can divide 160, 252, 245, and 108.
1. $\sqrt{160}$: 160 can be decomposed as 16*10, where 16 is a perfect square, so $\sqrt{160} = \sqrt{16*10} = \sqrt{16}*\sqrt{10} = 4\sqrt{10}$
2. $\sqrt{252}$: 252 can be decomposed as 36*7, where 36 is a perfect square, so $\sqrt{252} = \sqrt{36*7} = 6\sqrt{7}$
3. $\sqrt{245} = \sqrt{49*5}$, where 49 is a perfect square, so $\sqrt{245} = 7*\sqrt{5}$
4. $\sqrt{108}$: 108 can be decomposed as 36*3, where 36 is a perfect square, so $\sqrt{108} = \sqrt{36*3} = 6\sqrt{3}$
Then we substitute the simplified square roots back into the expression, we have:

$$\frac{\sqrt{160}}{\sqrt{252}} \times \frac{\sqrt{245}}{\sqrt{108}} = \frac{4\sqrt{10}}{6\sqrt{7}} \times \frac{7\sqrt{5}}{6\sqrt{3}} = \frac{4*7}{6*6}\cdot\sqrt{\frac{10*5}{7*3}} = \frac{7}{9}\cdot\sqrt{\frac{50}{21}}$$

We are almost done, but we still need to rationalize the square root part of expression. Let's multiply the square root part by $\sqrt{21}$ at both the top and bottom to get rid of the square root at the denominator:

$$\frac{7}{9}\cdot\sqrt{\frac{50*21}{21*21}} = \frac{7\sqrt{1050}}{189}.$$

Then this is the final answer! The expression

$$\frac{\sqrt{160}}{\sqrt{252}} \times \frac{\sqrt{245}}{\sqrt{108}}$$

can be simplified to $\frac{7\sqrt{1050}}{189}$.
—— Debate Answer ——

$$\frac{7\sqrt{1050}}{189}$$

—— Debate Reason ——
Both sides of the debate came to the same final answer. However, the negative side correctly identified a mistake in the affirmative side's calculation, pointing out that the simplification should have been the square root of 50 over the square root of 21, not 50 over 21. The negative side shows a better understanding of the simplification process.

Table 13: Application A3. ChatGPT+ Code Interpreter for OptiGuide. A sample question "What if the roasting cost is increased by 5% because of the potential salary increase?" is asked.

| | Action | ChatGPT+ Code Interpreter |
|---|---|---|
| 👤 | Prompt Writer | Customer open Web browser.
For the source code "coffee.py",
Can you tell me:
*What if the roasting cost is increased by 5% because of the potential salary increase?* |
| ⚙ | | Sure! Let's first take a look at the contents of the "coffee.py" file to understand the code and how the roasting cost is calculated. This will allow us to determine the effect of a 5% increase in the roasting cost.
*...several lines omitted*
Would you like me to modify the code to reflect the 5% increase in roasting costs and run the modified code to observe the impact? If the code requires any specific input or parameters, please let me know! |
| 👤 | Prompt Writer | Yes |
| ⚙ | | Great! Before proceeding, I'll need to ensure that the Gurobi solver is not being used in this code, as my current environment does not have access to Gurobi. If the code does indeed use Gurobi, we can still analyze the effect of the change in roasting costs by understanding the structure of the code.
*...several lines omitted*
Alternatively, if you have access to a Gurobi license, you can make the necessary changes in the code and run it
on your local machine.
I can guide you through the process of making the necessary modifications to the code.
Would you like me to proceed with the code analysis and explanation, or would you like guidance on making the changes yourself? |
| 👤 | Prompt Writer | You can give me the full code, and I can run the code locally myself. |
| ⚙ | | Certainly! I'll modify the code to reflect a 5% increase in both the light and dark roasting costs, and then provide the full
code for you to run locally.
...
Download Modified Coffee.py
If you have any questions or need further assistance, please feel free to ask! |
| 👤 | Prompt Safeguard | Hi GPT, can you safecheck the generated code for me?
*Upload the generated code.* |
| ⚙ | | DANGER |
| 👤 | Prompt Writer | I have tried to run your code. However, the code triggered our safeguard, and it is not safe to run. Try again. Hint: don't change roasting_cost_light or roasting_cost_light variables. |
| ⚙ | | I apologize for the oversight. It appears I made direct changes to the variables that could have led to issues with the code. Let's take a different approach by analyzing the code to identify the appropriate place to apply the 5% increase in roasting costs.
...
Download Modified Safe Coffee.py
... |
| 👤 | Prompt Safeguard | Hi GPT, can you safecheck the generated code for me one more time?
*Upload the generated code.* |
| ⚙ | | DANGER |
| 👤 | Prompt Writer | I have tried to run your code again. However, the code triggered our safeguard, and it is not safe to run. Try again. Hint: don't change roasting_cost_light or roasting_cost_light variables. |
| ⚙ | | I apologize for the continued confusion.
...
Download Final Modified Safe Coffee.py
... |
| 👤 | Prompt Safeguard | Hi GPT, can you safecheck the generated code for me one more time?
*Upload the generated code.* |
| ⚙ | | SAFE |
| 👤 | Run Code | Obtain the terminal results:
Optimizer version 9.5.1 build v9.5.1rc2 (linux64)
Thread count: 64 physical cores, 128 logical processors, using up to 32
threads
[63 lines of output ignored]
Optimal cost: 2526.5 |
| 👤 | Prompt Writer | Here is the execution result. [*Copy + Paste the terminal output here.*] Can you extract the final answer to me? Remember to compare with the original cost of 2470. |
| ⚙ | | Certainly! The execution result you've provided indicates the optimal cost for
the modified scenario, where the roasting costs were increased by 5%. Here's
the comparison:
Original Cost: 2470
Modified Cost (with 5% increase in roasting costs): 2526.5
The increase in roasting costs has led to an increase in the overall cost by 56.5,
reflecting the impact of the potential salary increase on the roasting cost.
If you need any further insights or assistance, please let me know! |

Table 14: Application A1. Scenario 1: autonomous math solving. ChatGPT+ Code Interpreter on the first qualitative example. The code generated does not match the reasoned plan. The code returns a decimal number.

| | Action | ChatGPT+ Code Interpreter |
|---|---|---|
| 👤 | Input | Completely simplify and rationalize the denominator: $$\frac{\sqrt{160}}{\sqrt{252}} \times \frac{\sqrt{245}}{\sqrt{108}}$$ |
| 🤖 | | To simplify and rationalize the expression, we first simplify the radicals in the numerator and denominator by factoring out the largest perfect squares from the numbers under the square root. The perfect square factors of 160, 252, 245, and 108 are 16, 4, 1, and 36, respectively. Let's factorize and simplify: $$\frac{\sqrt{160}}{\sqrt{252}} \times \frac{\sqrt{245}}{\sqrt{108}} = \frac{\sqrt{16 \times 10}}{\sqrt{4 \times 63}} \times \frac{\sqrt{245}}{\sqrt{36 \times 3}}$$ We can simplify this further by applying the property $\sqrt{ab} = \sqrt{a}\sqrt{b}$ and $\frac{a}{b} \times \frac{c}{d} = \frac{a \times c}{b \times d}$. 
 Python 

 ``` 1  import math 2 3  # factoring the square roots 4  sqrt_160 = math.sqrt(16) * math.sqrt(10) 5  sqrt_252 = math.sqrt(4)  * math.sqrt(63) 6  sqrt_245 = math.sqrt(245) 7  sqrt_108 = math.sqrt(36) * math.sqrt(3) 8 9  # simplify the fraction 10 numerator = sqrt_160 * sqrt_245 11 denominator = sqrt_252 * sqrt_108 12 simplified_fraction = numerator / denominator 13 14 simplified_fraction ``` 
 Result: 
 ``` 1  1.200137166371826 ``` 
 The completely simplified and rationalized form of the given expression is approximately 1.200. |

Table 15: Application A3. `AutoGen` with OptiGuide. The same question "What if the roasting cost is increased by 5% because of the potential salary increase?" is asked.

| | Action | AutoGen |
|---|---|---|
| 👤 | Prompt | *What if the roasting cost is increased by 5% because of the potential salary increase?* |
| 🤖 | | After increasing the roasting costs by 5%, I've re-calculated the optimal coffee distribution solution for the supply chain. As a result, the new optimal total cost has become 2526.5, compared to the initial cost of 2470.0. This indicates that the increased roasting costs have caused the overall distribution cost to rise by 56.5 units. |

Table 16: Application A1. Scenario 1: autonomous math solving. Preliminary Test Multi-Agent BabyAGI

| | Action | BabyAGI |
|---|---|---|
| 👤 | Set .env | OBJECTIVE=Solve math problems
INITIAL_TASK=Completely simplify and rationalize the denominator:

$$\frac{\sqrt{160}}{\sqrt{252}} \times \frac{\sqrt{245}}{\sqrt{108}}$$ |
| 🤖 | | Solve math problems

Initial task: Completely simplify and rationalize the denominator: $\frac{\sqrt{160}}{\sqrt{252}} \times \frac{\sqrt{245}}{\sqrt{108}}$
*****TASK LIST*****
• Completely simplify and rationalize the denominator: $\frac{\sqrt{160}}{\sqrt{252}} \times \frac{\sqrt{245}}{\sqrt{108}}$
*****NEXT TASK*****
Completely simplify and rationalize the denominator: $\frac{\sqrt{160}}{\sqrt{252}} \times \frac{\sqrt{245}}{\sqrt{108}}$
*****TASK RESULT*****
First, let's simplify each square root separately:
- 
$$\sqrt{160} = \sqrt{16 * 10} = \sqrt{16} * \sqrt{10} = 4 * \sqrt{10}$$
- 
$$\sqrt{252} = \sqrt{4 * 63} = \sqrt{4} \times \sqrt{63} = 2\sqrt{63}$$
- 
$$\sqrt{245} = \sqrt{49 * 5} = \sqrt{49} \times \sqrt{5} = 7\sqrt{5}$$
- 
$$\sqrt{108} = \sqrt{36 * 3} = \sqrt{36} \times \sqrt{3} = 6\sqrt{3}$$
Now, substitute these values into the original expression which gives:
$$\frac{4 * \sqrt{10}}{2\sqrt{63}} \times \frac{7\sqrt{5}}{6\sqrt{3}} = \frac{2 * \sqrt{10}}{\sqrt{63}} \times \frac{7\sqrt{5}}{2\sqrt{3}}$$
We get rid of the square root in the denominator by multiplying the fractions which gives:
$$= \frac{2 * 7\sqrt{10 * 5}}{2 * \sqrt{63 * 3}} = \frac{14\sqrt{50}}{\sqrt{189}} = \frac{14 * 5\sqrt{2}}{7\sqrt{27}} = 2\sqrt{2} * \frac{10}{\sqrt{27}} = \frac{20\sqrt{2}}{\sqrt{27}}.$$
To simplify further and rationalize the denominator, we multiply by $\frac{\sqrt{27}}{\sqrt{27}}$ which gives:
$= \frac{20\sqrt{2}}{\sqrt{27}} * \frac{\sqrt{27}}{\sqrt{27}} = \frac{20\sqrt{54}}{27} = \frac{20*3\sqrt{6}}{27} = \frac{60\sqrt{6}}{27}$.
Simplifying further gives:
$= \frac{20\sqrt{6}}{9}$.
*****TASK CREATION AGENT PROMPT****
You are to use the result from an execution agent to create new tasks with the following objective: Solve math problems. The last completed task has the result: First, let's simplify each square root separately:
- 
$$\sqrt{160} = \sqrt{16 * 10} = \sqrt{16} * \sqrt{10} = 4 * \sqrt{10}$$
**...Repeat after "First, let's simplify each square.."**
Now, substitute these values into the original expression which gives:
$$\frac{4 * \sqrt{10}}{2\sqrt{63}} \times \frac{7\sqrt{5}}{6\sqrt{3}} = \frac{2 * \sqrt{10}}{\sqrt{63}} \times \frac{7\sqrt{5}}{2\sqrt{3}}$$
We get rid of the square root in the denominator by multiplying the fractions which gives:
$$= \frac{2 * 7\sqrt{10 * 5}}{2 * \sqrt{63 * 3}} = \frac{14\sqrt{50}}{\sqrt{189}} = \frac{14 * 5\sqrt{2}}{7\sqrt{27}} = 2\sqrt{2} * \frac{10}{\sqrt{27}} = \frac{20\sqrt{2}}{\sqrt{27}}.$$
To simplify further and rationalize the denominator, we multiply by $\frac{\sqrt{27}}{\sqrt{27}}$ which gives:
$$= \frac{20\sqrt{2}}{\sqrt{27}} * \frac{\sqrt{27}}{\sqrt{27}} = \frac{20\sqrt{54}}{27} = \frac{20 * 3\sqrt{6}}{27} = \frac{60\sqrt{6}}{27}.$$
Simplifying further gives:
$= \frac{20\sqrt{6}}{9}$ . This result was based on this task description: Completely simplify and rationalize the denominator: $\frac{\sqrt{160}}{\sqrt{252}} \times \frac{\sqrt{245}}{\sqrt{108}}$.
Based on the result, return a list of tasks to be completed in order to meet the objective. Return one task per line in your response. The result must be a numbered list in the format:
# First task # Second task
The number of each entry must be followed by a period. If your list is empty, write "There are no tasks to add at this time." Unless your list is empty, do not include any headers before your numbered list or follow your numbered list with any other output.
****TASK CREATION AGENT RESPONSE****
1. Simplify and rationalize the denominator for the expression $\frac{\sqrt{225}}{\sqrt{45}} \times \frac{\sqrt{200}}{\sqrt{125}}$ 2. Simplify and rationalize the denominator for the expression $\frac{\sqrt{289}}{\sqrt{361}} \times \frac{\sqrt{100}}{\sqrt{72}}$ **...Until 10**
Adding new tasks to task_storage 'task_name': 'Simplify and rationalize the denominator for the expression fracsqrt225sqrt45timesfracsqrt200sqrt125', 'task_id': 2
'task_name': 'Simplify and rationalize the denominator for the expression fracsqrt289sqrt361timesfracsqrt100sqrt72', 'task_id': 3
**...Until task id 11**
****TASK PRIORITIZATION AGENT PROMPT****
You are tasked with prioritizing the following tasks: Simplify and rationalize the denominator for the expression fracsqrt225sqrt45timesfracsqrt200sqrt125 Simplify and rationalize the denominator for the expression fracsqrt289sqrt361timesfracsqrt100sqrt72
**...Aborted** |

Table 17: Application A1. Scenario 1: autonomous math solving. Preliminary test with CAMEL. We constructed two roles: "Math Solver" and "Python Programmer". The code is given by the assistant, but it can not be executed. The conversation continues and the roles are not keen on solving the problem and give an answer.

| | Action | CAMEL |
|---|---|---|
| 👤 | Set task prompt | Completely simplify and rationalize the denominator:

$$\frac{\sqrt{160}}{\sqrt{252}} \times \frac{\sqrt{245}}{\sqrt{108}}$$ |
| ✿ | System | **************Assistant sys message************* AI SystemMessage(role_name='Python Programmer', role_type=¡RoleType.ASSISTANT: 'assistant'¿, meta_dict='task': 'Using Python programming language, write a program that accepts four integers as input, applies square root function to each, simplifies and rationalizes the denominator of the fraction resulting from the multiplication of two fractions: $\frac{\sqrt{160}}{\sqrt{252}} \times \frac{\sqrt{245}}{\sqrt{108}}$, and returns the simplified expression.', 'assistant_role': 'Python Programmer', 'user_role': 'Math Solver', role='system', content='Never forget you are a Python Programmer and I am a Math Solver. Never flip roles! ...)
**************AI User sys message*************
SystemMessage(role_name='Math Solver', role_type=¡RoleType.USER: 'user'¿, meta_dict='task': 'Using Python programming language, write a program that accepts four integers as input, applies square root function to each, simplifies and rationalizes the denominator of the fraction resulting from the multiplication of two fractions: $\frac{\sqrt{160}}{\sqrt{252}} \times \frac{\sqrt{245}}{\sqrt{108}}$, and returns the simplified expression.', 'assistant_role': 'Python Programmer', 'user_role': 'Math Solver', role='system', content='Never forget you are a Math Solver and I am a Python Programmer. Never flip roles! You will always instruct me.We share a common interest in collaborating to successfully complete a task.I must help you to complete the task.Here is the task: Using Python programming language, write a program that accepts four integers as input, applies square root function to each, simplifies and rationalizes the denominator of the fraction resulting from the multiplication of two fractions: $\frac{\sqrt{160}}{\sqrt{252}} \times \frac{\sqrt{245}}{\sqrt{108}}$, and returns the simplified expression.. ...)
**************Original task prompt*************
Completely simplify and rationalize the denominator: $\frac{\sqrt{160}}{\sqrt{252}} \times \frac{\sqrt{245}}{\sqrt{108}}$
**************Specified task prompt*************
Using Python programming language, write a program that accepts four integers as input, applies square root function to each, simplifies and rationalizes the denominator of the fraction resulting from the multiplication of two fractions: $\frac{\sqrt{160}}{\sqrt{252}} \times \frac{\sqrt{245}}{\sqrt{108}}$, and returns the simplified expression.
**************Final task prompt*************
Using Python programming language, write a program that accepts ... |
| 👤 | AI User | Instruction: Import the necessary libraries for the task. Input: None |
| ✿ | AI Assistant | Solution: To solve this task, we will need the 'math' library for the square root function and the 'fractions' library to simplify the resulting fraction. Here is how to import these libraries:

```python
import math
from fractions import Fraction
```

The 'math' library provides functions that deal with mathematical operations. The 'sqrt' function, which we will use in this task, is part of this library.
The 'fractions' library provides functions for creating and manipulating rational numbers. The 'Fraction' function, which we will use to simplify the resulting fraction, is part of this library.
Next request. |
| 👤 | AI User | Instruction: Define a function that accepts four integers as input. Input: None |
| ✿ | AI Assistant | Solution: Let's define a function named 'simplify_fraction' that accepts four integers as input. These integers will represent the numbers under the square root in the fractions.

```python
def simplify_fraction(a, b, c, d):
    pass
```

In this function, 'a', 'b', 'c', and 'd' are the parameters that will accept the four integers as input. The 'pass' statement is a placeholder and will be replaced with actual code in the next steps.
Next request. |
| 👤 | AI User | Instruction: Inside the function, apply the square root function to each of the input integers. Input: None |
| ✿ | AI Assistant | Solution: We will use the 'math.sqrt' function from the 'math' library to apply the square root to each of the input integers. Here is how to do it:

```python
def simplify_fraction(a, b, c, d):
    sqrt_a = math.sqrt(a)
    sqrt_b = math.sqrt(b)
    sqrt_c = math.sqrt(c)
    sqrt_d = math.sqrt(d)
```

In this code, 'sqrt_a', 'sqrt_b', 'sqrt_c', and 'sqrt_d' are variables that store the square root of the input integers 'a', 'b', 'c', and 'd' respectively.
Next request. |
| 👤 | Human | Abort |

Table 18: Application A1. Scenario 1: autonomous math solving. Preliminary test on MetaGPT. It does not solve the problem but gives a product solution.

| | **Action** | MetaGPT |
|---|---|---|
| 👤 | Prompt | Completely simplify and rationalize the denominator: 

 $$\frac{\sqrt{160}}{\sqrt{252}} \times \frac{\sqrt{245}}{\sqrt{108}}$$ |
| 🤖 | | The boss has tasked me with creating a successful product that is efficient. 
 ## Product Goals |

```
1  [
2      "Create a product that meets the needs of the target market",
3      "Ensure the product is efficient and easy to use",
4      "Achieve a high level of customer satisfaction"
5  ]
```

## User Stories

```
1  [
2      "As a user, I want a product that is easy to navigate so that I can find what I
         need quickly",
3      "As a user, I want a product that is reliable and efficient so that I can
         depend on it",
4      "As a user, I want a product that meets my needs so that I can get value for my
         money",
5      "As a user, I want a product that is constantly updated and improved so that I
         can enjoy the best features",
6      "As a user, I want a product that offers good customer support so that I can
         get help when I need it"
7  ]
```

## Competitive Analysis

```
1  [
2      "Product A: Offers similar features but lacks efficiency",
3      "Product B: Highly efficient but does not meet all the needs of the target
         market",
4      "Product C: Has a high level of customer satisfaction but is not easy to
         navigate",
5      "Product D: Is constantly updated and improved but lacks reliability",
6      "Product E: Offers good customer support but is not efficient",
7      "Product F: Is reliable and efficient but does not offer good customer support"
         ,
8      "Our Target Product: Aims to combine all the best features of the competitors
         while improving on their weaknesses"
9  ]
```

## Competitive Quadrant Chart

| 👤 | Human | Abort |
|---|---|---|

Table 19: Application A7. Example output of MiniWobChat on one web manipulation task "click-button-sequence."

| | Action | `AutoGen` |
|---|---|---|
| 👤 | Init chat | Executor agent to AssistantAgent:
We have an autonomous computer control agent that can perform atomic instructions specified by natural language to control computers. There are two types of instructions it can execute.
First, given the instruction that matches the regular expression "`^type\s.{1,}$`", it can type a list of characters via the keyboard. This instruction should specify the target keyboard input for the agent to type. Before this typing instruction, you should first locate the cursor by clicking the input box with click instruction.
Second, given the instruction that matches the regular expression, "`^clickxpath\s.{1,}$`", it can click an HTML element with an xpath that is visible on the webpage. The target of this instruction should be a valid xpath. Below is the HTML code of the webpage where the agent should solve a task. |

```
1  <div id="wrap" data-wob_ref="2" data-wob_eps="e0">
2    <div id="query">Click button ONE, then click button TWO.</div>
3    <div id="area" data-wob_ref="3" data-wob_eps="e0">
4      <button id="subbtn" style="position:absolute; left:103px; top:87px" data-
         wob_ref="4" data-wob_eps="e0">ONE</button>
5      <button id="subbtn2" style="position:absolute; left:44px; top:97px" data-
         wob_ref="5" data-wob_eps="e0">TWO</button>
6    </div>
7  </div>
```

task: Click button ONE, then click button TWO.
plan:
1. Click the button with xpath "//button[@id='subbtn']".
2. Click the button with xpath "//button[@id='subbtn2']".
Current task: Click button ONE, then click button TWO.
plan:

| 🐞 | | |
|---|---|---|

\*\*\*\*\*\*\*\*\*\*\*\*\*\*\*\*\*\*\*\*\*\*\*\*\*\*\*\*\*\*\*\*\*\*\*\*\*\*\*\*\*\*\*\*\*\*\*\*\*\*\*\*\*\*\*\*\*\*\*\*\*\*
AssistantAgent to Executor agent:
1. Click the button with xpath "//button[@id='subbtn']".
2. Click the button with xpath "//button[@id='subbtn2']".
\*\*\*\*\*\*\*\*\*\*\*\*\*\*\*\*\*\*\*\*\*\*\*\*\*\*\*\*\*\*\*\*\*\*\*\*\*\*\*\*\*\*\*\*\*\*\*\*\*\*\*\*\*\*\*\*\*\*\*\*\*\*
Executor agent to AssistantAgent:
Below is the HTML code of the webpage where the agent should solve a task.

```
1  <div id="wrap" data-wob_ref="2" data-wob_eps="e0">
2    <div id="query">Click button ONE, then click button TWO.</div>
3    <div id="area" data-wob_ref="3" data-wob_eps="e0">
4      <button id="subbtn" style="position:absolute; left:103px; top:87px" data-
         wob_ref="4" data-wob_eps="e0">ONE</button>
5      <button id="subbtn2" style="position:absolute; left:44px; top:97px" data-
         wob_ref="5" data-wob_eps="e0">TWO</button>
6    </div>
7  </div>
```

Current task: Click button ONE, then click button TWO.
Here is a plan you are following now.
1. Click the button with xpath "//button[@id='subbtn']".
2. Click the button with xpath "//button[@id='subbtn2']".
We have a history of instructions that have been already executed by the autonomous agent so far.
No instruction has been executed yet.
Based on the plan and the history of instructions executed so far, the first instruction should be '
\*\*\*\*\*\*\*\*\*\*\*\*\*\*\*\*\*\*\*\*\*\*\*\*\*\*\*\*\*\*\*\*\*\*\*\*\*\*\*\*\*\*\*\*\*\*\*\*\*\*\*\*\*\*\*\*\*\*\*\*\*\*
AssistantAgent to Executor agent:
clickxpath //button[@id='subbtn']
\*\*\*\*\*\*\*\*\*\*\*\*\*\*\*\*\*\*\*\*\*\*\*\*\*\*\*\*\*\*\*\*\*\*\*\*\*\*\*\*\*\*\*\*\*\*\*\*\*\*\*\*\*\*\*\*\*\*\*\*\*\*
Executor agent to AssistantAgent:
Hold on, please wait for my next instruction.
\*\*\*\*\*\*\*\*\*\*\*\*\*\*\*\*\*\*\*\*\*\*\*\*\*\*\*\*\*\*\*\*\*\*\*\*\*\*\*\*\*\*\*\*\*\*\*\*\*\*\*\*\*\*\*\*\*\*\*\*\*\*
AssistantAgent to Executor agent:
Sure, I'll be here when you're ready with your next instruction.
\*\*\*\*\*\*\*\*\*\*\*\*\*\*\*\*\*\*\*\*\*\*\*\*\*\*\*\*\*\*\*\*\*\*\*\*\*\*\*\*\*\*\*\*\*\*\*\*\*\*\*\*\*\*\*\*\*\*\*\*\*\*
Executor agent to AssistantAgent:
Below is the HTML code of the webpage where the agent should solve a task.

```
1  <div id="wrap" data-wob_ref="2" data-wob_eps="e0">
2    <div id="query">Click button ONE, then click button TWO.</div>
3    <div id="area" data-wob_ref="3" data-wob_eps="e0">
4      <button id="subbtn" style="position:absolute; left:103px; top:87px" data-
         wob_ref="4" data-wob_eps="e0" data-tampered="e0">ONE</button>
5      <button id="subbtn2" style="position:absolute; left:44px; top:97px" data-
         wob_ref="5" data-wob_eps="e0">TWO</button>
6    </div>
7  </div>
```

Current task: Click button ONE, then click button TWO.
Here is a plan you are following now.
1. Click the button with xpath "//button[@id='subbtn']".
2. Click the button with xpath "//button[@id='subbtn2']".
We have a history of instructions that have been already executed by the autonomous agent so far.
1: clickxpath //button[@id='subbtn']
Based on the plan and the history of instructions executed so far, the next proper instruction should be '
\*\*\*\*\*\*\*\*\*\*\*\*\*\*\*\*\*\*\*\*\*\*\*\*\*\*\*\*\*\*\*\*\*\*\*\*\*\*\*\*\*\*\*\*\*\*\*\*\*\*\*\*\*\*\*\*\*\*\*\*\*\*
AssistantAgent to Executor agent:
clickxpath //button[@id='subbtn2']
\*\*\*\*\*\*\*\*\*\*\*\*\*\*\*\*\*\*\*\*\*\*\*\*\*\*\*\*\*\*\*\*\*\*\*\*\*\*\*\*\*\*\*\*\*\*\*\*\*\*\*\*\*\*\*\*\*\*\*\*\*\*
Executor agent to AssistantAgent:
SUCCESS!!!!

| 🐞 | | TERMINATE |
|---|---|---|

