# OpenReview forum: "AutoGen: Enabling Next-Gen LLM Applications via Multi-Agent Conversation"
_ICLR.cc/2024/Conference — Submitted to ICLR 2024_

### Official Review · Reviewer_rXeX · 2023-10-31

**Soundness:** 3 good
**Presentation:** 3 good
**Contribution:** 2 fair
**Rating:** 6
**Confidence:** 4

**Summary:**

The paper introduces AutoGen to enhance the development of LLM applications through the use of multiple conversational agents. These agents are customizable, capable of various modes of operation, and can interact with each other, human inputs, and tools to accomplish complex tasks. The framework supports programming interaction behaviors using both natural language and code, catering to a wide range of applications across different domains.
Contributions:
- The framework provides a generic and extensible design for agents, enabling them to engage in conversations and multi-turn interactions seamlessly.
- AutoGen introduces a conversation-centric programming paradigm, simplifying the development of complex LLM applications and providing adaptability to a wide range of needs.

**Strengths:**

- AutoGen’s conversation-centric programming paradigm can simplify and unify the development of complex LLM applications, showcasing originality in application workflow design.
- AutoGen provides robust support for developers, including the ability to program agent interactions using both natural language and code, catering to a diverse set of development preferences and needs.

**Weaknesses:**

While AutoGen presents a promising framework for multi-agent applications of LLMs, the paper tends to read more like a tech report rather than a traditional research paper, as it lacks a focused exploration of research-oriented problems. The novelty of the work seems constrained by this, as it primarily introduces and elaborates on the framework’s capabilities without diving deep into scientific inquiries or hypotheses testing. Although the practical utility of AutoGen is clear, the analysis provided in the paper regarding its positioning and performance relative to existing solutions is not sufficiently comprehensive, and the discussion on scalability, performance overheads, and real-world applications remains superficial. The agent customization, while a strong feature, could benefit from clearer guidelines, and the human-AI interaction aspect requires more elaboration. These areas of improvement highlight a need for a more detailed, research-focused approach, and suggest that the work may find a more fitting audience at a venue like the System Demonstration track at ACL, where applied tools and frameworks are showcased and appreciated.

**Questions:**

- Is there any scenarios in which the single agents outperform the multi-agent counterparts? For example, the Alfworld may be as easy as possible for a single agent to solve, maybe the assistant agent and executor agent can be the same one which generates the formatted [think + act] steps at the same time, which makes the multi-agent problem into a prompt template design problem.
- How did the grounding agent in the A3 of Figure 4 know the crucial commonsense knowledge, Did you design the specific prompts/few-shot examples to teach it?
- Why do only two methods have the whole dataset results in A1 of Figure 4?
- In the scenario of the A6 of Figure 3, multi-agent players are playing chess, how to make sure every agent strictly follows the chess rules?

---

> ### Author Response · Authors · 2023-11-21
> **Authors' responses for Reviewer rXeX. Thank you for your comments!**
>
> We thank the reviewer for the comments and questions! Please find our responses below. We would like to answer any further questions you might have!
>
> ## On weakness about venue fit
> AutoGen is motivated by the emerging need of multi-agent systems based on LLM, tools, and human input, and the challenges around building such systems. Given the focus on facilitating the development of multi-agent systems, we selected “infrastructure, software libraries, hardware, etc.” as the primary area of this submission.
>
> Our contributions/novelty lies in:
> (1) the concepts/techniques we proposed for easily creating and orchestrating multi-agent systems, e.g., conversable agent, conversation programming, and conversation-centric way of building multi-agent systems.
> (2) the experiments we conducted to quantify the value of multi-agent settings.
> (3) Additionally, we open-source the library to encourage additional research and development to explore and accelerate progress in this emerging area.
>
> ## Re Question 1 on single-agent vs multi-agent performance
> In our experiments, we observed that the multiagent settings are comparable to or better than the single-agent counterparts. The cases where multi-agent settings do not provide additional value are typically cases where the tasks are easier and can be solved reliably with a single agent.
>
> For example, we see that on ALFWorld (Figure 4.c), a two-agent setup is comparable to a single-agent (with ReAct). However, a three-agent setup (with a grounding agent) outperforms both the React agent and the 2-agent setup. We note the performance of multi-agent settings will vary depending on the agent design. Hence, it is conceivable that a badly designed multi-agent solution could underperform a single agent.
>
> ## Re Question 2 on the grounding agent in ALFWorld
>
> We hard-coded a set of common sense knowledge relevant to household tasks in the grounding agent in A4. E.g., “You must find and take the object before you can examine it.” as shown in Figure 10 in Appendix D. The knowledge will be requested if the assistant’s proposed solution fails three times or the task gets stuck. No example or prompt is used to teach it.
>
> ## Re Question 3 on the evaluation of A1 in Figure 4
> Note that the whole MATH dataset includes 5000 questions. Running each method on the dataset costs between $500 and $800. Some methods (that use ChatGPT Plus) require substantial manual effort and are also restricted by message and token hourly rates - see footnote 4 on page 7.
>
> Our experiments using the level-5 problems (the smaller dataset)  indicated a significant performance gap between the methods AutoGen, LangChain, and ReAct and that GPT-4 performs slightly better. As such, we prioritized comparing AutoGen to GPT-4 on the larger (5000 questions) dataset.
>
> ## Re Question 4 on how to make agents follow chess rules
> To ensure adherence to chess rules in our multi-agent chess game, we have introduced a 'Board Agent' that leverages the Python chess library for move validation.
> This agent operates by extracting the UCI (Universal Chess Interface) move from each player's response at every turn. It then verifies the legality of this move using the chess library. Should a move be deemed illegal, the Board Agent issues an error message to the respective player, requesting a new legal move. For an illustrative example of the Board Agent's functionality, both with and without its intervention, please refer to Figure 16 in Appendix D, located on page 31 of our submission.

---

> > ### Author Response · Authors · 2023-11-23
> >
> > Dear Reviewer rXeX,
> >
> > Thank you again for your valuable comments on our submission. In response to your comments, we have provided detailed responses. We are writing to follow up and inquire if our responses have adequately answered your questions and addressed the venue fit issue you raised. If so, we respectfully request that you consider raising the rating of our work. We are fully open to any further elaboration or clarification you might need to facilitate this reevaluation.  We greatly appreciate your time and effort in reviewing our work and look forward to your re-evaluation. Thank you!
> >
> > Sincerely,
> > Authors of this submission

---

### Official Review · Reviewer_xV6g · 2023-10-31

**Soundness:** 2 fair
**Presentation:** 1 poor
**Contribution:** 1 poor
**Rating:** 5
**Confidence:** 3

**Summary:**

This paper proposes an open-source toolkit AutoGen, It introduces a tool to facilitate developing multi-agent LLM backed systems. The tool considers three different backend handlers for requests including LLM agent, human and some tools like code executor. It provides developers an opportunity to use both natural language or code for interaction. It covers 6 different use cases and shows promising results against default GPT benchmarks and some other tools.

**Strengths:**

+ Open-source multi-agent programming framework is definitely an interesting project.
+ Authors include some empirical results to showcase their performance of the tools.
+ Authors provide extensive documentation to explain different use cases.

**Weaknesses:**

- Not much takeaway in terms of scientific learning.
I don’t find as a reader what lessons or results we can get from the paper. The main message is that we have a tool that can help developing multi-agent conversation. In my personal opinion, developing it based on a mature LLM agent is neither time-consuming nor scientifically challenging.

- The results are not convincing.
I found the comparison of results is not rigorously evaluated and not convincing. For example, the paper shows that by repeating the question again to LLM may potentially get better results. But the results are not convincing due to few sample they use nor making a lot of sense.

- Too much brag about their system but little evidence is shown to support it.
For example, in the introduction, they aim to design a “capable, reusable, customizable, and effective” system. I didn’t see support towards it.

**Questions:**

Can you specify why the interactive retrival can lead to better results? I am curious about what the result will look like if I try different variations of questions? As with interactive, I can think of the improvement is due to more trials other than better problem understanding.

---

> ### Author Response · Authors · 2023-11-21
> **Authors' responses to Reviewer xV6g (part 1). Thank you for your comments!**
>
> We thank the reviewer's comments and questions. Please find our responses below. We are happy to answer any more questions the reviewer might have.
>
> ## (Re Weakness 1) On scientific contribution and practical utility
> > Comment: Not much takeaway in terms of scientific learning. I don’t find as a reader what lessons or results we can get from the paper. The main message is that we have a tool that can help developing multi-agent conversation.
>
> In addition to displaying engineering excellence, we believe AutoGen furthers science on multi-agents in the context of LLMs in many ways.
> - The concepts of conversable agents and conversation programming are novel and help systematically understand and make progress on the nascent topics of multi-agent for LLM applications. For example, as highlighted by reviewer SJWG, , “Another useful insight of this paper is to divide the main application workflow into small multi-agent conversations…”
> - Empirical evaluation in Section 3 and in Appendix D shows AutoGen's advantages over existing frameworks and also produces some new discoveries. For example, through the experiments introduced in A4, we discover that “a multi-agent design is helpful in boosting performance in coding tasks that need safeguards”. We provide further elaboration on the takeaways from this and other applications in Appendix D (Application Details).
> - The proposed concepts and abstractions combined with our OSS implementation enable other researchers to explore multi-agents and further science. E.g, as mentioned in A4 of Section 3 in the main paper, through AutoGen OptiGuide’s implementation was reduced from over 430 lines to just 100 lines. This is a clear testament to the framework's ability to facilitate more efficient scientific experiments.
>
> > Comment: The main message is that we have a tool that can help developing multi-agent conversation. In my personal opinion, developing it based on a mature LLM agent is neither time-consuming nor scientifically challenging.
>
> Although the reviewer does not personally perceive the framework’s value in supporting the development of multi-agent systems, the framework is well-received by the open-source community in general:  as of 20th November 2023, our OSS implementation
> - has been downloaded over 130,000 times within less than 2 months (we omit package name to remain anonymous.);
> - has been used to build many applications from the open-source community (used in more than 200 open-source projects). For example, GPT academic, an LLM-based service for assisting various academic-related activities such as scientific writing, research paper reading, etc,; and
> - has been awarded one of the top 100 open-source projects 2022-2023 by the International Open Benchmark Council under the following evaluation criteria:
>     - The key milestones that lay the foundation for open source movement or development;
>    - The original or pioneering open source works;
>    - The open source works that play a significant role in promoting the development of software and hardware;
>    - The open source works are widely used or cited by industry or academia.
>
> These are all evidence of our framework's practical utility for AI applications and experiments. Although we totally understand and value the reviewer's personal preference, we hope the reviewer can take the general impact and utility this work could bring into consideration.

---

> ### Author Response · Authors · 2023-11-21
> **Authors' responses to Reviewer xV6g (part 2). Thank you for your comments!**
>
> ## (Re weakness 2) On evaluation and results
>
> Thank you for your feedback. We understand the importance of rigorous evaluation in assessing the effectiveness of our framework.
> To ensure a comprehensive understanding, our study encompasses a blend of qualitative and quantitative evaluations across a variety of applications. These evaluations are designed to provide a holistic view of the framework's performance and utility.
>
> > I found the comparison of results is not rigorously evaluated and not convincing.
>
> Since different evaluation types may yield varying insights, we would like to ask the reviewer which application and evaluation this comment is referring to here.
>
> To further clarify, as mentioned in the paper, the evaluation of all four applications (A1-A4) with quantitative evaluation results followed established practices from existing literature. E.g.,
>
> - For A1, our evaluation used both the full MATH dataset [1], which includes 5000 problem instances, and a sub-sampled set including 120 problems.
> - For A2, our evaluation was performed on all the problem instances from the Natural Questions benchmark [2], which is a benchmark for Question Answering Research and includes 6775 question instances. We followed the evaluation procedure established in [3].
> - For A3, our evaluation is performed on the ALFWorld dataset [4], following the evaluation procedure used in the ReAct [5] paper.
> - For A4, our evaluation is performed following the evaluation procedure established in the original OptiGuide paper [6].
>
> In addition to quantitative evaluation results, we also include a rich set of qualitative studies for A1-A4 (included in Appendix D Application Details).
>
> A5 and A6 are innovative applications for demonstration purposes (as there is no relevant benchmark for these two tasks) and thus are primarily supported by qualitative.
>
> [1] Hendrycks, Dan, et al. "Measuring mathematical problem solving with the math dataset." NeurIPS 2021.
>
> [2] Kwiatkowski, Tom, et al. "Natural questions: a benchmark for question answering research." ACL 2019.
>
> [3] Adlakha, Vaibhav, et al. "Evaluating correctness and faithfulness of instruction-following models for question answering." arXiv preprint arXiv:2307.16877 (2023).
>
> [4] Shridhar, Mohit, et al. "Alfworld: Aligning text and embodied environments for interactive learning." ICLR 2021
>
> [5] Yao, Shunyu, et al. "React: Synergizing reasoning and acting in language models." ICLR 2023.
>
> [6] Li, Beibin, et al. "Large language models for supply chain optimization." arXiv preprint arXiv:2307.03875 (2023).
>
> ## (Re weakness 3) On the support of claims
>
> We presented numerous applications developed with the proposed framework and provided very comprehensive evaluations of most of the applications in both Section 2 and Appendix D (from page 19 to page 43). Those are all evidence supporting the advantages of the proposed framework.  Regarding the example mentioned by the review on “capable, reusable, customizable, and effective”, we do provide support for this claim:
>
> - Capable: In the second paragraph of Section 2.1, we introduced 'Agent capabilities powered by LLMs, humans, and tools', elaborating the supported agent capabilities.
> - Customizable: Section 2.1 discusses 'Agent customization and cooperation'. In the applications presented in Section 3, we developed agents customized from the built-in agent. This includes the Retrieval-augmented User Proxy agent in Application 2 and the ALF World Executor agent in Application 3.
> - Reusable: The built-in agent, Assistant, is reused in Applications 1 (including two scenarios), 2 (including two scenarios), and 4.
> - Effective: In Applications 1-4, we demonstrate how AutoGen enables the development of multi-agent systems that are effective in solving tasks such as math problem-solving, retrieval augmented question answering, decision-making, and coding with safety constraints.

---

> ### Author Response · Authors · 2023-11-21
> **Authors' responses to Reviewer xV6g (part 3). Thank you for your comments!**
>
> ## Re question on interactive retrieval
> We thank the reviewer for the insightful question! We agree with the reviewer’s point that the improved performance with interactive retrieval ultimately stems from the additional trials requesting context. However, achieving this wisely and robustly is highly non-trivial and does require a proper understanding of the problem status.
>
> To better understand the challenge here, let’s consider a naive approach: this approach performs multiple trials of retrieval right at the beginning when the question is asked. This gives us more trials but does not require problem understanding at all. However, this naive approach has one obvious limitation: it is hard to decide how many up-front trials one should try to include. Including context from too many trials can lead to the conversation exceeding the context length and may incur unnecessarily high inference cost. When the multiple up-front trials are conducted separately, one also needs a way to select the answer from it, which is also non-trivial. Conversely, too few trials may not provide adequate context. This latter case with a single trial is investigated in our ablation study in A2, with results illustrated in Figure 4(b) and Figure 10 in Appendix D. Regarding the former case, since it relies on the number of trials as a hyperparameter and an answer selection method, it is easy to construct scenarios that have undesirable results, e.g., failure due to context overflow or incurring a very high cost.  Interactive retrieval, on the other hand, can be considered an online approach that naturally remedies this tension: It does not rely on a pre-specified trial number but makes the “UPDATE CONTEXT” request when necessary (based on LLM’s understanding of whether the question can or cannot be answered without further context) as the conversation proceed until the question is considered answered.
>
> We would like to hear if your comments are addressed, and the question is answered, and we would be happy to provide further elaboration if needed.

---

> ### Author Response · Authors · 2023-11-23
>
> Dear Reviewer xV6g,
>
> Thank you again for your valuable comments on our submission. In response to your comments, we have provided detailed responses. We are writing to follow up and inquire if our responses have adequately answered your question on interactive retrieval and addressed your concerns about the weaknesses of this work. If so, we respectfully request that you consider raising the rating of our work. We are fully open to any further elaboration or clarification you might need to facilitate this reevaluation. We greatly appreciate your time and effort in reviewing our work and look forward to your re-evaluation. Thank you and we look forward to hearing from you.
>
> Sincerely,
>
> Authors of this submission

---

### Official Review · Reviewer_AgUb · 2023-11-01

**Soundness:** 3 good
**Presentation:** 3 good
**Contribution:** 3 good
**Rating:** 8
**Confidence:** 3

**Summary:**

This paper presents AutoGen, an open-source framework for building LLM applications via multi-agent conversations. The paper introduces two key concepts: conversable agents and conversation programming. Conversable agents are entities that can communicate with each other and have different capabilities powered by LLMs, humans, or tools. Conversation programming is a paradigm that allows developers to define the interaction behavior between agents using a fusion of natural and programming languages. The paper demonstrates six applications of AutoGen that span various domains and complexities and shows that AutoGen can achieve better performance, reduce development effort, and enable novel LLM usage scenarios.

**Strengths:**

(1)	The proposed framework simplifies the overall complex LLM workflows and enables automation. By using conversable and customizable agents, it supports conversational modes for complex workflows. It provides a collection of working systems with different complexities. These systems cover a wide range of applications from various domains and complexities.

(2)	The proposed method allows developers to use a fusion of natural and programming languages to define agent behaviors and conversation patterns.

(3)	The paper demonstrates the effectiveness and generality of the framework in various domains and tasks. It showcases novel and innovative applications that are enabled by the multi-agent conversation framework.

**Weaknesses:**

(1)	The paper does not address the issue of context length, which may become too long as the number of conversation turns increases. This could affect the performance and efficiency of the LLMs and the agents.


(2)	It would be better to consider the cost issue, which is important for practical applications. The experiments are conducted on GPT-4 and GPT-3.5, which are expensive and not widely accessible. How would the framework perform on open-source LLMs with lower capacity?


(3)	In my opinion, AutoGen appears to be an extension of CAMEL, both supporting agent role-playing and agent conversations. The authors discuss the related work of CAMEL in the appendix, and highlight two distinct advantages of AutoGen: 1) its capability for tool usage and 2) dynamic conversation. However, it deserves to give an in-depth discussion about CAMEL and AutoGen about their differences in the introduction section. Actually, I do not think the tool-usage is a big challenge if using the GPT4. It is expected to discuss the challenge from static conversation (of CAMEL) to dynamic conversation (of AutoGen).

**Questions:**

(1)	Please refer to Weakness (2). How would the framework perform on open-source LLMs with lower capacity?

(2)	Please refer to Weakness (3). What is the technique challenge for introducing dynamic conversation compared with AutoGen?

(3)	Can this method be applied in other complex tasks, such as automatically using professional tools (Oracle, MATLAB)?

---

> ### Author Response · Authors · 2023-11-21
> **Authors' responses to Reviewer AgUb (part 1). Thank you for your comments!**
>
> We thank the reviewer for the insightful comments, constructive suggestions, and thoughtful questions!
>
> ## (Re weakness 1) On context length
> Thanks to the extensibility of our framework, we are able to work out two solutions to address this potential issue.
> We are able to implement a `CompressibleAgent` under the proposed AutoGen framework that could compress long context when needed. Please find [a notebook example demonstrating how to use CompressibleAgent in this anonymous link](https://anonymous.4open.science/r/autogen-iclr2024/notebook/agentchat_compression.ipynb).
>
> As mentioned in the expanded discussion section in Appendix B, we acknowledge the existence of scenarios/cases where other libraries/packages could help. The context length problem could be one such case. AutoGen has been integrated with MemGPT, which is a recent work that could teach LLMs to manage memory for unbounded context. Please find a notebook example [demonstrating how AutoGen can be used together with MemGPT](https://anonymous.4open.science/r/autogen-iclr2024/notebook/memgpt_coder_autogen.ipynb).
>
> ##  (Re question 1) On performance with open-source LLMs
> In general, open-source LLMs with lower capacity than GPT-3.5/4 would lead to lower performance if not meticulously utilized. Fortunately, our multi-agent framework provides the flexibility to use those open-source models in strategic ways.  In one of our follow-up work (we do not disclose the title of work due to anonymity concerns), we built a two-agent system using the built-in AssistantAgent and UserProxyAgent in AutoGen and evaluated the system on tasks [1] that require coding and external API calls. In this application, if we directly replace GPT-3.5/4 with LLAMA-2-13B-chat in the AssistantAgent, the system’s performance indeed drops by a large margin. However, we also show that by including multiple AssistantAgent backed by GPT models or LLAMA-2-13B-chat into a multi-agent system, we can actually reduce the dollar cost of the system while improving the success rate.
>
> ----
>
> | AssistantAgent     | Success Rate (averaged over 300 queries) | Average Cost
> |---------|----------|----------|
> | LLAMA-2-13B-chat  | 13.33% | 0.00
> | GPT-3.5-turbo | 34.11% | $0.41
> | GPT-4   | 77.22% | $13.93
> | GPT-4 +  GPT-3.5-turbo +  LLAMA-2-13B-chat  | 83.22% |  $8.98
>
> [1] ToolLLM: Facilitating Large Language Models to Master 16000+ Real-world APIs

---

> ### Author Response · Authors · 2023-11-21
> **Authors' responses to Reviewer AgUb (part 2). Thank you for your comments!**
>
> ## (Re question 2) On technical challenges
>
> We thank the reviewer for the constructive suggestion and this insightful question.
>
> We would like to first clarify that CAMEL has a different focus and positioning from AutoGen: CAMEL is primarily positioned as a framework for studying the cooperative behaviors of agents of different roles, not an infrastructure to support the development of LLM applications as the case in AutoGen. This difference can be better understood from the two frameworks’ different behaviors in solving a task. E.g., under the task “*Design a custom game using pygame*” [demonstrated in CAMEL’s official GitHub repo](https://colab.research.google.com/drive/1AzP33O8rnMW__7ocWJhVBXjKziJXPtim?usp=sharing#scrollTo=-IhYiAWDme66), we compared AutoGen vs. CAMEL and summarized in results in this [anonymized document](https://anonymous.4open.science/r/autogen-iclr2024/notebook/alternatives/README.md).  From the comparison, we can see that CAMEL is primarily simulating a conversation between an AI agent with the role “*Computer Programmer*", and an agent with the role “*gamer*”, but is not actually creating a meaningful game; while AutoGen is able to actually create games with pygame and save the created game into a file such that it can be directly executed and played.
>
> One more fundamental distinction, which poses profound technical challenges, is AutoGen’s general support for multi-agent systems with an agent number N > 2. CAMEL’s inception prompting based role-playing framework currently primarily supports systems with two AI agents with potentially a critic in the loop. There is no general support for systems with more than 2 agents or with other conversation patterns.
>
> Note that moving from 2 to N (N > 2) in an effective way that could support LLM applications is highly non-trivial and is technically challenging: When N = 2, the communication between the agents is straightforward. Supporting N > 2, in general, requires careful abstraction and implementation so that the framework can (1) possess the flexibility to meet various application needs (there is hardly a one-fit-all pattern), and (2) support conversation patterns that can make meaningful progress in task completion. AutoGen is so far the only framework that realizes both objectives decently well.
>
> We will clarify the major differences and challenges in the main paper accordingly in a future draft.
>
> ## (Re question 3) On the support of complex tasks and professional tools.
>
> Yes, our framework is adaptable to other complex tasks involving the use of professional tools. In the Multi-agent Coding application introduced in A4, in addition to Python, the developed OptiGuide system uses several professional tools, including Gurobi for mathematical optimization used by the commander agent, and various tools in Docker used by the AssistantAgent.
> In general, tools that can be accessed using Python code can be seamlessly incorporated into function calls, and can therefore be automatically used by agents. Both MATLAB and Oracle have officially supported Python APIs and could be supported.

---

> ### Author Response · Authors · 2023-11-23
>
> Dear reviewer AgUb,
>
> Thank you again for your encouraging and constructive comments, as well as the insightful questions regarding our submission. We hope our responses, together with the additional empirical results, have addressed your questions and concerns on the potential weaknesses. If so, we respectfully request that you consider raising the rating of our work in light of the added empirical results and additional support we provide for addressing the context length issue, which is highly non-trivial.
>
> We greatly appreciate your time and effort in reviewing our work.
>
> Best,
>
>
> The authors of this submission

---

### Official Review · Reviewer_SJWG · 2023-11-08

**Soundness:** 3 good
**Presentation:** 3 good
**Contribution:** 3 good
**Rating:** 8
**Confidence:** 2

**Summary:**

The authors present an open-source framework to develop LLM applications using multiple agents that interact with each other to complete tasks. The agents presented are customizable, can converse with each other and can operate in various modes using LLM, human input or tools. Through experiments, they show the effectiveness of this framework for several tasks like  Math Problem Solving, Question-Answering task etc.

**Strengths:**

-	The approach defines a generic design for agents that can use LLMs, human inputs, certain tools or combination of them. LLM agents can use capabilities such as role playing, progress making from conversation history, proving feedback. Human involvement can be configured at different levels e.g. frequency and conditions for when to request human input. Tools agents can execute code/functions (suggested by LLMs).
Combining these agents in different configurations can result in powerful agents with very different capabilities.

-	The another useful insight of this paper is to divide the main application workflow into small multi-agent conversations which they call Conversation programming. It consists of two components: computation which is the actions that the agents take to get their response, control-flow which defines the decisions on which agents to send messages to. These two components allow for control over the conversation flow in the application workflow.

**Weaknesses:**

None

**Questions:**

-	In Figure 4(c), the performance of ReAct on Best of 3 is better than AutoGen (2 agents). I am curious as to what do you think would the reason for that?

---

> ### Author Response · Authors · 2023-11-21
> **Authors' responses to Reviewer SJWG. Thank you for your comments!**
>
> We thank the reviewer for the comments and the question. The difference between the performance of ReAct and AutoGen (2 agents) is actually quite marginal. The marginal difference could be caused by randomness from LLM output generation.

---

> ### Author Response · Authors · 2023-11-23
>
> Dear reviewer SJWG,
>
> Thank you again for your encouraging comments, insightfully pointing out the strengths of this work. We are writing to follow up if your question on the performance difference between ReAct and AutoGen 2-agent has been addressed by our reponse. Please feel free to let us know if you need further clarification or elaboration.
>
> We highly appreciate your time and effort spent on reviewing our paper!
>
> Sincerely,
>
> The authors of this submission

---

### Public Comment · ~Guohao_Li1 · 2023-11-13
**Missing references and comparison with existing work**

The paper presents AutoGen, an open-source framework for creating Large Language Model (LLM) applications through multi-agent conversations. AutoGen's agents are customizable, conversable, and can operate in various modes, employing combinations of LLMs, human inputs, and tools. However, the paper lacks a comprehensive comparison with related work in their main paper, particularly in discussing the propsed concepts of AutoGen over existing open source multi-agent frameworks [1][2].

**Conceptual Similarity:** Both AutoGen and CAMEL focus on multi-agent systems for LLM applications, emphasizing the autonomous cooperation of agents in task-solving.

**Agent Roles:** AutoGen and other agent frameworks share similar concepts of having distinct roles for agents. The *AssistantAgent* and *UserProxyAgent* in AutoGen closely resemble the *AI Assistant* and *AI User* in CAMEL [1], where agents are assigned distinct roles and collaborate interactively to achieve specific tasks. *Critic/human-in-the-loop*​​ are also dicussed in both AutoGen and CAMEL. Moreover. the *GroupChatManager* proposed in AutoGen is not a novel concept which has been explored in many open-source projeccts such as LangChain [2]. In LangChain's Multi-agent authoritarian speaker selection example, a *DirectorDialogueAgent* is used to dynamically select next-speaker.

**Notable Omission:** AutoGen does not properly cite or compare its framework with CAMEL, LangChain and other open-source projects in the main paper, despite the conceptual similarities and potential overlaps in the application of multi-agent systems for LLM applications. This omission could be considered a significant gap in acknowledging related work in the field.

**Summary**: The paper presents a valuable open-source framework, which shows practical utility in various applications. However, the lack of demonstrated novelty and insufficient engagement with existing literature, particularly in not citing or comparing with relevant works like CAMEL [1], LangChain [2] and other open-source frameworks in the main paper, limits its contribution as a research paper. The current form of AutoGen appears to be an amalgamation of existing techniques, yet it does not adequately acknowledge or credit prior related work in the main paper. The paper would benefit from a clearer articulation of its unique scientific contributions and a more thorough comparison with existing frameworks in the main paper. Therefore, I suggest a more robust positioning of AutoGen within the context of existing research would be beneficial for the research and open-source community.

[1] Li, Guohao, Hasan Abed Al Kader Hammoud, Hani Itani, Dmitrii Khizbullin, and Bernard Ghanem. "CAMEL: Communicative Agents for" Mind" Exploration of Large Language Model Society." NeurIPS 2023. https://openreview.net/forum?id=3IyL2XWDkG. https://github.com/camel-ai/camel.

[2] Harrison Chase. Langchain. 2022. Multi-agent authoritarian speaker selection by Michael Chang. https://github.com/langchain-ai/langchain/blob/master/cookbook/multiagent_authoritarian.ipynb

---

> ### Author Response · Authors · 2023-11-22
>
> We would like to thank the author of CAMEL for taking the time to comment on our paper.
>
> **Concerning the Alleged Notable Omission**: We respectfully disagree with the commenter’s opinion on notable omission.
>
> As clarified in the main paper, CAMEL, LangChain Agents (and other recent LLM-based multi-agent framework/systems) are not published at peer-reviewed venues by the time this work is submitted and shall be considered contemporaneous work. We thus do not have the obligation to cite or compare with them. We would like to refer to [ICLR 2024 Reviewer Guide](https://iclr.cc/Conferences/2024/ReviewerGuide) (the last question in "FAQ for reviewers" section) for official guidance on this matter.
>
> > “Q: Are authors expected to cite and compare with very recent work? What about non peer-reviewed (e.g., ArXiv) papers? (updated on 7 November 2022)"
>
> > "A: We consider papers contemporaneous if they are published (available in online proceedings) within the last four months. That means, since our full paper deadline is September 28, if a paper was published (i.e., at a peer-reviewed venue) on or after May 28, 2023, **authors are not required to compare their own work to that paper**…
> ”
>
> Despite the fact that we do not have the obligation to do so, we did make a significant effort to acknowledge the relevant contemporaneous work:
>
> 1. We cited those contemporaneous work (see the reference list on pages 11-13 right after the main paper);
> 2. We provided a discussion of those contemporaneous work in Appendix A;
> 3. We did compare with those contemporaneous work: see our comparison with multi-agent debate in Figure 4(a) and the comparison with multi-agent BabyAGI in Table 16, with CAMEL in Table 17, and with MetaGPT in Table 18. We only include multi-agent debate in the quantitive comparison in the main paper mainly because the other contemporaneous work is either not applicable or not very competitive on the scenarios we evaluated based on our pilot qualitative study (results included in the appendix).
>
> **Regarding the Claim of Conceptual Similarity in Agent Roles**: We also disagree with the commenter's assessment of conceptual similarity in terms of agent roles.
>
> First, the key concept about agents introduced by our work is the “*customizable and conversable agents*,” as clearly stated in the introduction section (the first bullet point on page 2) and Section 2.1. Different from CAMEL, in AutoGen, we managed to support LLMs, tool, human, and their mixture all as agents. Developers could compose a customized conversable agent depending on their actual needs. As introduced in the second paragraph of Section 2.1, the `AssistantAgent`, `UserProxyAgent ` and  `GroupChatManager ` are just three pre-configured customizations of the fundamental “ConversableAgent”. As presented in Section 3 Application, one can conveniently construct other customized agents, e.g., the “SafeGuard” agent,  “Chess Board” agent, “ALF Executor” agent based on the `ConversableAgent,` that possess different capabilities.
>
> Second, `UserProxyAgent` in AutoGen significantly differs from the AI User in CAMEL: it is designed to act as a human proxy for soliciting human inputs and performing tool execution, in contrast to the AI-based nature of the AI User in CAMEL. The so-called “similarity” is only on the surface level in terms of the general “user” and “assistant” roles, which are also commonly used in inference APIs of most chat-optimized LLMs (e.g., GPT-3.5, GPT-4).
>
> **Clarification on the unique contributions**:
> Finally, we do not agree with the commenter’s notion that AutoGen is “an amalgamation of existing techniques”. The conversable agent and conversation programming abstractions are novel. The novel abstraction and effective implementation together make the framework versatile and flexible to elegantly support diverse applications of different complexities, both reflected by the demonstrated applications in this paper (together with the rich empirical studies) and the open-source projects that already benefit from AutoGen.
>
> However, based on these discussions, we will make the scientific contribution clearer in the paper. In particular, we will update Section 4 of the paper.
>
> Thank you again for your time and attention.

---

> ### Public Comment · ~Victor_Escorcia1 · 2023-11-26
> **Unethical behavior in this submission. Call the PC &/OR rigor police!**
>
> I personally worked with Guahao and other CAMEL authors. They are nice and polite researchers, rather than privilege unethical employees of a top institution in AI. I'll be surprised if anyone prove that they're acting unethically by requesting a proper citation and discussion of CAMEL in the main paper.
>
> Let's use our ethical and scientist hats,
> 1. CAMEL was accepted to NeurISP2023, right?
> 2. This ICLR 2024 submission knew about it.
> 3. Why isn't CAMEL discussed in the related work?
>
> I beg the AC to trigger the ethical committee of the conference to clarify the dispute. Justice without reparation & retribution are symptoms of unhealthy communities.
>
> P.D. 🤞🏼 authors do not compromise the brand of... let say **Microsoft** or any of the top frontier companies in AI.

---

### Meta-Review · Area_Chair_6Jmg · 2023-12-06

**Metareview:**

Overall, the paper presents a useful technique to combining multiple LLMs, and provides a software tool that is likely to be of interest. In these, it seems above the bar for acceptance.

However, there are issues which make me uncertain about this paper.

R4 raises important questions about how the approach posed in this paper is studied. There are important differences between insightful research and tech reports. This is crisply illustrated by naming the experimental section "applications". This section is presenting compelling results on interesting domain/problems, but lacks a methodological study in the sense of ablations, learning curves, etc. This diminishes what a researcher can learn from this paper and what this paper contributes to our shared body of knowledge.

There's a big issue of contextualization in existing work with this paper, especially with regards to CAMEL (Li et al. 2023 -- appeared on arXiv March, 2023). There's no discussion of this work in the main paper. The author did discuss it in the appendix, but this is a broken practice, because the supplementary material is not up for reviewing at the same level as the paper. Simply put, this seems to allow authors to say they cite and discuss papers, while not actually doing it in the part of the paper that 99% of the people are likely to read. I find this practice concerning. That said, according to ICLR guidelines, arXiv papers don't count as prior work until published in a peer review conference. While I understand the rationale behind this policy, in the current climate of publication, it can lead to weird outcomes. Because of ICLR's policy, I am not going to factor this into my decision, but it decreases my confidence.

In writing this meta review, I didn't take into account the public comments posted on the discussion board.

Because of the scientific deficiencies of the experiments, and after much consideration, I am going to recommend to reject the paper. But I do value two of the reviewers really like the paper, and I do see the value of the work.

**Justification For Why Not Higher Score:**

See above.

**Justification For Why Not Lower Score:**

See above.

---

### Decision · Program_Chairs · 2024-01-16

Reject